

# Long-term study of cloud radiative effect, cloud fraction and cloud type at two stations in Switzerland using hemispherical sky cameras

Christine Aebi[1,2], Julian Gröbner[1], Niklaus Kämpfer[2], and Laurent Vuilleumier[3]

[1]Physikalisch-Meteorologisches Observatorium Davos, World Radiation Center, Davos, Switzerland.
[2]Oeschger Center for Climate Change Research and Institute of Applied Physics, University of Bern, Bern, Switzerland.
[3]Federal Office of Meteorology and Climatology MeteoSwiss, Payerne, Switzerland.

*Correspondence to:* Aebi Christine (christine.aebi@pmodwrc.ch)

**Abstract.** The current study analyses time series of cloud radiative effect during daytime depending on cloud fraction and cloud type at two stations in Switzerland. Information about fractional cloud coverage and cloud type is retrieved from images taken by visible all-sky cameras. The longwave cloud radiative effect (LCE) for low-level clouds and a cloud coverage of 8 octas has a median value between 57 and 71 $\mathrm{Wm}^{-2}$. For mid- and high-level clouds the LCE is significantly lower. It is shown that the fractional cloud coverage, the cloud base height (CBH) and integrated water vapour (IWV) all have an influence on the magnitude of the LCE. These observed dependences have also been modelled with the radiative transfer model MODTRAN5. The relative values of the shortwave cloud radiative effect (SCE$_{rel}$) for low-level clouds and a cloud coverage of 8 octas are between -88 to -61 %. Also here the higher the cloud is, the less negative the SCE$_{rel}$ values are. In cases where the measured direct radiation value is below the threshold of 120 $\mathrm{Wm}^{-2}$ the SCE$_{rel}$ decreases substantially, while cases where the measured direct radiation value is larger than 120 $\mathrm{Wm}^{-2}$ lead to a SCE$_{rel}$ of around 0 %. In 13 % and 8 % of the cases in Davos and Payerne respectively a cloud enhancement has been observed with a maximum in the cloud class cirrocumulus-altocumulus at both stations.

## 1 Introduction

The influence of clouds on the radiation budget and radiative transfer of energy in the atmosphere persist the greatest sources of uncertainty in the simulation of climate change (*Boucher et al.*, 2013). Small changes in cloudiness and radiation can have large impacts on the Earth's climate. There are two competing influences of clouds on the surface radiation budget (*Sohn and Bennartz*, 2008): On one hand, clouds reflect incoming shortwave radiation and thus diminish the incoming energy on the Earth's surface. On the other hand, they prevent longwave radiation from the surface and lower atmosphere from escaping the atmosphere. Radiation is the energy source which modifies the atmospheric thermodynamic structure, the Earth's general circulation and the climate system (*Sohn and Bennartz*, 2008). The effect of clouds is not only of importance in the long term temporal and spatial averages but also on shorter timescales (seconds to minutes). Furthermore, the exchange of energy due to the formation of clouds and precipitation is an important component of the global water cycle and in turn of climate change (*Trenberth*, 2011). Thus the influence of clouds has to be measured and analysed in more detail.

Not only the cloud amount but also other cloud parameters such as e.g. cloud type and cloud optical thickness are of impor-





tance. The physical parameters defining the various cloud types may have distinct effects on radiation of different wavelengths. For example optically thin and high-level clouds have a relatively small effect on the downward shortwave radiation, whereas low-level and thick clouds scatter and absorb a large part of the solar radiation and re-emit it as thermal radiation in all directions. Thus cloud type variations can alter both shortwave and longwave radiation fluxes due to changes in cloud levels, water

content and cloud temperatures (*Chen et al.*, 2000; *Allan*, 2011). However, not only different cloud types, but also clouds of the same type may have a distinct influence on the surface radiation budget due to their macrophysical (cloud coverage and geometry) and microphysical properties (e.g. optical thickness and particle size distribution) (*Pfister et al.*, 2003). The distribution, frequency and length of occurrence of different cloud types, and the cloud amount in general, may cause a change in climate variations and climate feedback (*Bony et al.*, 2006; *Norris et al.*, 2016). In order to assess the cloud climate feedback,

also cloud independent parameters such as time of year or time of day are of importance (*Allan*, 2011). Knowledge about the cloud type also allows conclusions to be drawn regarding the current atmospheric motions (*Chen et al.*, 2000). Thus additional information about the cloud type is crucial to categorize the cloud radiative effect (*Futyan et al.*, 2005).

In detailed numerical weather and climate prediction models, cloud properties (cloud base height, cloud cover and cloud thickness) and the physical processes responsible for the formation and dissipation of clouds are often approximations and

parametrisations (e.g. *Bony et al.* (2006); *Allan et al.* (2007); *Zelinka et al.* (2014); *Sherwood et al.* (2015)). In order to contribute to the accuracy of the representation of clouds in atmospheric prediction models, there is need for satellite and ground-based in situ measurements (*Sohn*, 1999; *Jensen et al.*, 2008; *Su et al.*, 2010; *Roesch et al.*, 2011). Satellite measurements have the advantage of covering a broader area. Mainly over the oceans it is almost the only data source to obtain information about cloud coverage and cloud type (*Ohring et al.*, 2005). From the Meteosat Second Generation (MSG) geostationary satellites,

data about clouds are taken with a time resolution of 15 minutes (*Werkmeister et al.*, 2015). A better temporal resolution may be obtained by the measurement of clouds with e.g. ground-based all-sky cameras.

For several years, all-sky cloud cameras have been in use world-wide in order to collect continuous information about clouds from the surface. Many studies already determined cloud coverage based on all-sky camera images (e.g. *Long et al.* (2006), *Kazantzidis et al.* (2012), *Alonso et al.* (2014)). *Heinle et al.* (2010) presented a method for using all-sky camera images to

classify cloud types. *Wacker et al.* (2015) applied, with slight modifications, this algorithm to determine six cloud classes automatically with a mean success rate of 50 to 70 %. The current study uses the cloud type detection and the cloud fraction algorithm presented in *Wacker et al.* (2015).

The current study presents time series of cloud radiative effects at the surface depending on cloud fraction and cloud types at two stations in Switzerland. The data and methods (including the description of the algorithms and the models) are described

in section 2. The time series of the cloud radiative effect in the longwave and shortwave regions at the two stations Davos and Payerne and their analyses are presented and discussed in section 3. Conclusions are outlined in section 4.



## 2   Data and Methods

### 2.1   Data

Data are available from two stations in Switzerland. The stations are located at two altitude levels, Payerne, located in the Midlands (46.49°N, 6.56°E, 490 m asl) and Davos, located in the Swiss Alps (46.81°N, 9.84°E, 1594 m asl). At both of these
stations a visible all-sky camera has been installed. The camera type in Payerne is a VIS-J1006, manufactured by Schreder GmbH (www.schreder-cms.com). This camera system consists of a commercial digital camera (Canon Power Shot A60) with a fisheye lens and a glass dome on top to protect the camera from rain and dust. This camera is sensitive in the red-green-blue (RGB) region of the spectrum and takes two images every five minutes with a resolution of $1200 \times 1600$ pixels each. The two images taken, one just after the other one, have different exposure times (1/500 s and 1/1600 s, respectively) but the same fixed
aperture of f/8.

The camera system in Davos is a Q24M from Mobotix (www.mobotix.com). It is a commercial surveillance camera with a fisheye lens sensitive in the RGB as well. The resolution of the images is the same as that for the camera in Payerne. In Davos, one image is taken every minute with an exposure time of 1/500 s. The Mobotix camera is ventilated and installed on a solar tracker with a shading disk.

The radiation data are retrieved from Kipp and Zonen CMP22 pyranometers (shortwave; 0.3 - 3 $\mu$m) and from Kipp and Zonen CG4 pyrgeometers (longwave; 3 - 100 $\mu$m) at both stations. All the instruments are daily cleaned and traceable to the World Radiation Center (WRC). The temperature data used in the current study are measured at 2 m height at both stations. The integrated water vapour (IWV) data are based on GPS measurements (*Bevis et al.*, 1992; *Hagemann et al.*, 2003) and retrieved from the STARTWAVE (STudies in Atmospheric Radiative Transfer and Water Vapour Effects) database (*Morland et al.*, 2006).
Aerosol optical depth data are retrieved from precision filter radiometers (PFR, manufactured by PMOD/WRC). Ceilometer data for the retrieval of the cloud base height (CBH) are only available in Payerne. At this station a CHM15k ceilometer from Jenoptik (now Lufft Mess- und Regeltechnik GmbH) is installed (*Wiegner and Geiß*, 2012).

For the Davos station a time series of the cloud radiative effect (CRE) has been calculated from August 7, 2013 to April 30, 2017 with a time resolution of one minute. For Payerne, the time series of the CRE includes data from January 1, 2013 to April
30, 2017 with a time resolution of five minutes. Data have only been taken into account for daytime measurements when the full sun disk is located above the horizon and the mountains. Cloud radiative effect data availability in these periods is around 98 % and 86 % for Davos and Payerne respectively which mainly results from occasional data gaps of 1 to 3 consecutive days. The lower data availability in Payerne can be explained by two longer time periods of more than 20 consecutive days when no camera data are available.


### 2.2   Cloud Radiative Effect

In the current study, the cloud radiative effect (CRE) is defined as a radiation measurement value minus a modelled clear sky value. The total cloud radiative effect (TCE) is divided into shortwave cloud radiative effect (SCE) and longwave cloud





radiative effect (LCE)

$$TCE = SCE + LCE \tag{1}$$

which are both calculated separately. For our calculations, only measurements from downward radiation during daytime are taken into account. In the following, the SCE values are given as relative values ($SCE_{rel}$) and calculated using Eq. 2.

$$SCE_{rel} = SCE / SCE_{CSM} * 100\% \tag{2}$$

where $SCE_{CSM}$ is the modelled clear sky irradiance value for the corresponding date and time. $SCE_{rel}$ is used due to the fact that different solar zenith angles lead to large differences in the absolute SCE values. Clouds usually increase the measured LW radiation at the surface as they emit LW radiation. Shortwave radiation measured at the surface is usually reduced by clouds as they reflect SW radiation back to space.

## 2.3 Clear Sky Models

For the calculation of the cloud radiative effects two clear sky models, one for the shortwave and the other one for the longwave region, are needed. The clear sky model for the longwave is an empirical model with input of measured surface temperature and integrated water vapour (IWV) values and a climatology of the atmospheric temperature profile (*Wacker et al.*, 2014). Comparing the LW radiation measurements of the clear sky cases, detected in the aforementioned time period, with the LW radiation values of the clear sky model gives a mean difference of $-0.8 \pm 3.9$ Wm$^{-2}$ and $-2.8 \pm 6.6$ Wm$^{-2}$ for Davos and Payerne respectively. Thus this difference lies within measurement uncertainty as it has also been shown by *Wacker et al.* (2014).

The shortwave clear sky model (used in Eq. 2) is a lookup table based on radiative transfer model calculations using Libradtran (*Mayer and Kylling*, 2005). The input of the model is a standard atmosphere including several measured atmospheric parameters: solar zenith angle (SZA), aerosol conditions (angstrom coefficient and aerosol optical depth (AOD), both interpolated over one day) and IWV. The airmass is calculated with the formula presented by *Kasten and Young* (1989). The lookup table is different for the two stations Davos and Payerne, considering a different range of values that might occur. Measured values of IWV, SZA and aerosol content are then interpolated with the lookup table and downward shortwave clear sky irradiance values are available for all the single time steps and the corresponding atmospheric conditions. The difference between SW measurement and the clear sky model is $4.7 \pm 19.6$ Wm$^{-2}$ ($0.5 \pm 6.2$ %) and $-0.5 \pm 45.8$ Wm$^{-2}$ ($-0.4 \pm 9.9$ %) for Davos and Payerne respectively.

## 2.4 Cloud Fraction and Cloud Type Retrievals

The calculation of the fractional cloud coverage is based on the all-sky cloud camera images from the aforementioned systems. Before calculating the cloud amount the images must be preprocessed. The distortion of the images has to be removed. Ad-





ditionally a horizon mask must be defined since Davos is located between two mountain ridges. For both stations the horizon mask has been defined on the basis of an individual clear sky image. After the preprocessing of the images a colour ratio (the sum of the blue to green ratio plus the blue to red ratio) is calculated per pixel (*Wacker et al.*, 2015). This calculated colour ratio is compared with a reference ratio value which is defined empirically in order to do the cloud classification per pixel. The

reference value for Davos is 2.2 and the one for Payerne 2.5. These values are different due to the differences in camera systems and settings. After comparing the calculated ratio with the reference value a decision can be made per pixel on a classification of cloud or clear sky. The fractional cloud coverage is then calculated as the sum of all cloudy pixels divided by the total number of sky pixels. For historical reasons the fractional cloud coverage is given in octas (*CIMO*, 2014). The classification of octas is taken from *Wacker et al.* (2015). Thus zero octa cloud coverage or cloud-free is defined as 0 - 5 % fractional cloud

coverage. Thus cloud-free does not necessarily mean no clouds at all. On the other end of the scale, eight octas is defined as a fractional cloud coverage of 95 % and more, which implies that it is not necessarily a fully covered sky. Octa 1 - 7 are defined in between with steps of 12.75 % fractional cloud coverage. For 65 - 85 % of the cases (in comparison to different cloud fraction retrieval instruments), the success rate of the fractional cloud cover calculation is $\pm 1$ octa (*Wacker et al.*, 2015).

The algorithm of *Heinle et al.* (2010) allows the classification of clouds based on statistical features retrieved from the all-sky

cloud images. This algorithm has been slightly adapted by *Wacker et al.* (2015) and is the one used for the current analysis. The classification is done by first calculating thirteen spectral, textural and radiative features. The features under consideration are the mean of the red and the mean of the blue channel, standard deviation and the skewness both of the blue channel, and the differences between the red and green, red and blue, and green and blue channels. The textural features are the energy, contrast and homogeneity of the blue channel and the total cloud coverage. The radiative feature longwave cloud radiative effect has

been added by *Wacker et al.* (2015) after testing its (positive) influence on the mean success rate of the cloud type recognition. The classifier used is the k-nearest-neighbour (knn) method, which is a supervised method. The training set to apply the knn method has been determined with visual analysis of the images. The training set is only available for the Payerne station. Thus, for both stations, Davos and Payerne, the same training set has been used. The training set contains only images with one cloud type present. However, the training images display a wide variety in the shape and position of the cloud. In the classification

procedure different cloud types per image might be detected, however as a result, only the one with the most hits is chosen. Thus only one cloud type per image is determined, although several might be present. The seven classes studied are cloud-free (Cf), cirrus-cirrostratus (Ci-Cs), cirrocumulus-altocumulus (Cc-Ac), stratocumulus (Sc), stratus-altostratus (St-As), cumulus (Cu) and cumulonimbus-nimbostratus (Cb-Ns). In the following, low-level clouds consist Cu, Sc, St-As and Cb-Ns. The cloud class Cc-Ac is a mid-level cloud class and Ci-Cs is a high-level cloud class.

According to *Wacker et al.* (2015), for a random data set of Davos, the situation Cf was correctly classified in more than 85 % of cases followed by Ci-Cs (65 %) and Cu (more than 50 % of the cases). For Payerne, around 80 % of the manually classified Sc clouds are also classified as such with the automatic algorithm and a random data set. The second most correctly detected cloud class is Cf (more than 70 % of the cases) and Cb-Ns (68 % of the cases). In the average, the success rates are 57 % and 55 % for Davos and Payerne respectively (*Wacker et al.*, 2015).






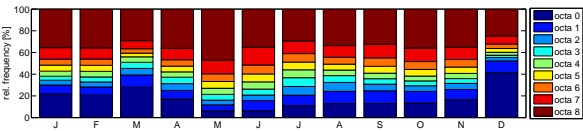 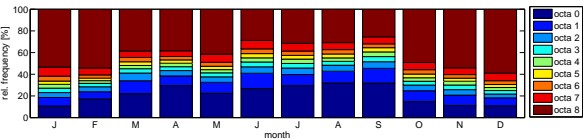

**Figure 1.** Relative frequencies of cloud coverages in 1 to 8 octa divisions (all cloud types together) for the two stations Davos (left) and Payerne (right).

## 3 Results and Discussion

### 3.1 Time Series of Cloud Radiative Effect

The data sets for the calculation of the cloud radiative effect (CRE) consist of 694,000 and 162,398 images for Davos and Payerne respectively. In Davos, the cloud coverage is eight octas for 34 % of the data set. In 17 % of the cases the cloud coverage is zero octa, which means a fractional cloud coverage of maximum 5 %. Seven octas cloud coverage occurs in 11 % of the cases followed by one octa (10 %). Two to six octas cloud coverage are all equally distributed in 5 to 6 % of the cases.

Also in Payerne, a cloud coverage of eight octas is determined in most of the cases (41 %), followed by zero octa in 22 % of the cases. In 10 % of the cases a cloud coverage of 1 octa is determined followed by seven octas (7 % of the cases) and two octas (5 %). A cloud coverage of three to six octas is determined in 3 - 4 % of the cases.

The distribution of the cloud coverage over the months is shown for Davos and Payerne separately in Figure 1. The colours indicate octa cloud coverages. In the winter half year (with a maximum in March and December) the sky is more often cloud-free than in the summer half year in Davos. In contrast, in May the sky is covered with eight octas in almost half of the cases. Cloud coverages of 1 to 7 octas are quite equally distributed over the months. In Payerne the situation is opposite for cloud-free days with more frequent eight octas cloud coverage in wintertime whereas cloud-free situations are more common during summertime. Also in Payerne, cloud coverages of 1 to 7 octas are fairly equally distributed.

Regarding the distribution of the cloud coverages in octas throughout the day, no real pattern can be observed in Davos. In Payerne there are more cloud-free conditions in the early morning than later in the day. The other octa cloud coverages are also equally distributed throughout the day.

In Davos, of the 694,000 cases, St-As, with 36 % of the cloud cases, is the cloud type that is most detected in the studied time period. The second and third most detected sky conditions in Davos are Cf and Cc-Ac with 17 % and 14 % respectively, followed by Sc (13 %), Cu (12 %), Ci-Cs (5 %) and Cb-Ns (3 %).

In Payerne, of the 162,398 sky images, in 28 % of the cases the cloud type Sc is detected. This is followed by Cf in around 22 % of cases, Cb-Ns (17 %), Cc-Ac (13 %), Ci-Cs (9 %), St-As (6 %) and Cu (4 %).

Figure 2 shows the relative frequencies of the cloud classes per month for the two stations Davos and Payerne separately and all cloud coverages together. In Davos, as determined by our algorithm, from October to May St-As is present in at least 40 % of the cases per month. The cloud class Cc-Ac is more often present in summertime than in wintertime. Ci-Cs is almost absent in the months August to October. This absence of the cloud class Ci-Cs in the late summer months does not match with the





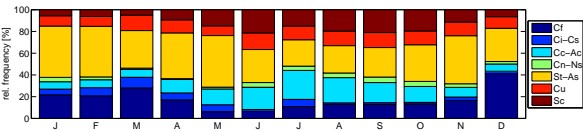 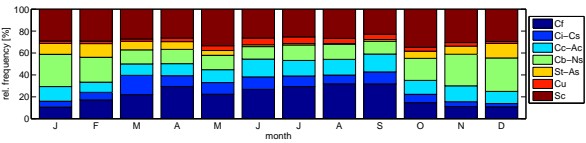

**Figure 2.** Relative frequencies of all cloud classes per month (all cloud coverages together) for the two stations Davos (left) and Payerne (right). Sc: stratocumulus, Cu: cumulus, St-As: stratus-altostratus, Cb-Ns: cumulonimbus-nimbostratus, Cc-Ac: cirrocumulus-altocumulus, Ci-Cs: cirrus-cirrostratus, Cf: cloud-free.

visual observation and might be explained by the fact that the camera system is not sensitive for the thin high-level clouds. The largest fraction of cloud type in Payerne is Sc for all months. The cloud classes Cb-Ns and St-As are both more often observed during wintertime than during summertime. The larger frequency of these two cloud types agree with the fact that there is more fully covered sky in wintertime than summertime.

Regarding the distribution of the cloud classes throughout the day, there are no large differences in the occurrence of cloud types per time of day. The distribution is quite flat for both stations.

### 3.1.1  Longwave Cloud Effect

As mentioned in section 2.2, the longwave cloud radiative effect (LCE) is calculated for Davos and Payerne and the six cloud
classes separately. The dependence of LCE on fractional cloud cover for the above mentioned time period for all six cloud classes is shown for Davos in Figure 3. The boxplots in the figure show the median (red line), the interquartile range (blue box) and the values that are within 1.5 times the interquartile range of the box edges (black line) per octa cloud coverage.

Figure 3 shows a non-linear increase in the LCE with increasing fractional cloud coverage for some cloud classes. This non-linear increase is clearly observed for the cumulus type clouds Cu, Sc and Cc-Ac. At present, it is not possible to explain this
non-linearity. The cloud classes St-As and Cb-Ns are mainly present with a cloud coverage of 5 octas and more. The LCE values for Ci-Cs in Davos and eight octas cloud coverage are clearly too high. Therefore no statistical values were calculated for this data set. Several reasons might be responsible for these too high values. One reason is that the camera is not sensitive enough to detect thin Ci-Cs clouds. Another reason might be that the cloud type algorithm is trained with a data set from Payerne, which may result in a misclassification of the thin and high-level clouds. That some data points are in the group of
Ci-Cs and eight octas cloud coverage is the result of a misclassification of the cloud type with the algorithm. These manually checked misclassified data points result in the detection of a weakness of the algorithm. The greater the fractional cloud coverage, the more difficult it becomes to distinguish among cloud types. This weakness can be explained by the fact that the greater the cloud coverage, the more difficult it becomes to distinguish textural features with the knn method and thus to distinguish among cloud types. For the cloud type Cc-Ac there are several LCE values of around 40 $\mathrm{Wm^{-2}}$ and small cloud coverages.
These high values are obtained in early mornings when the cloud is located in the vicinity of the horizon.

Table 2 gives an overview of the median values and their interquartile range of the LCE per octa cloud coverage for the six





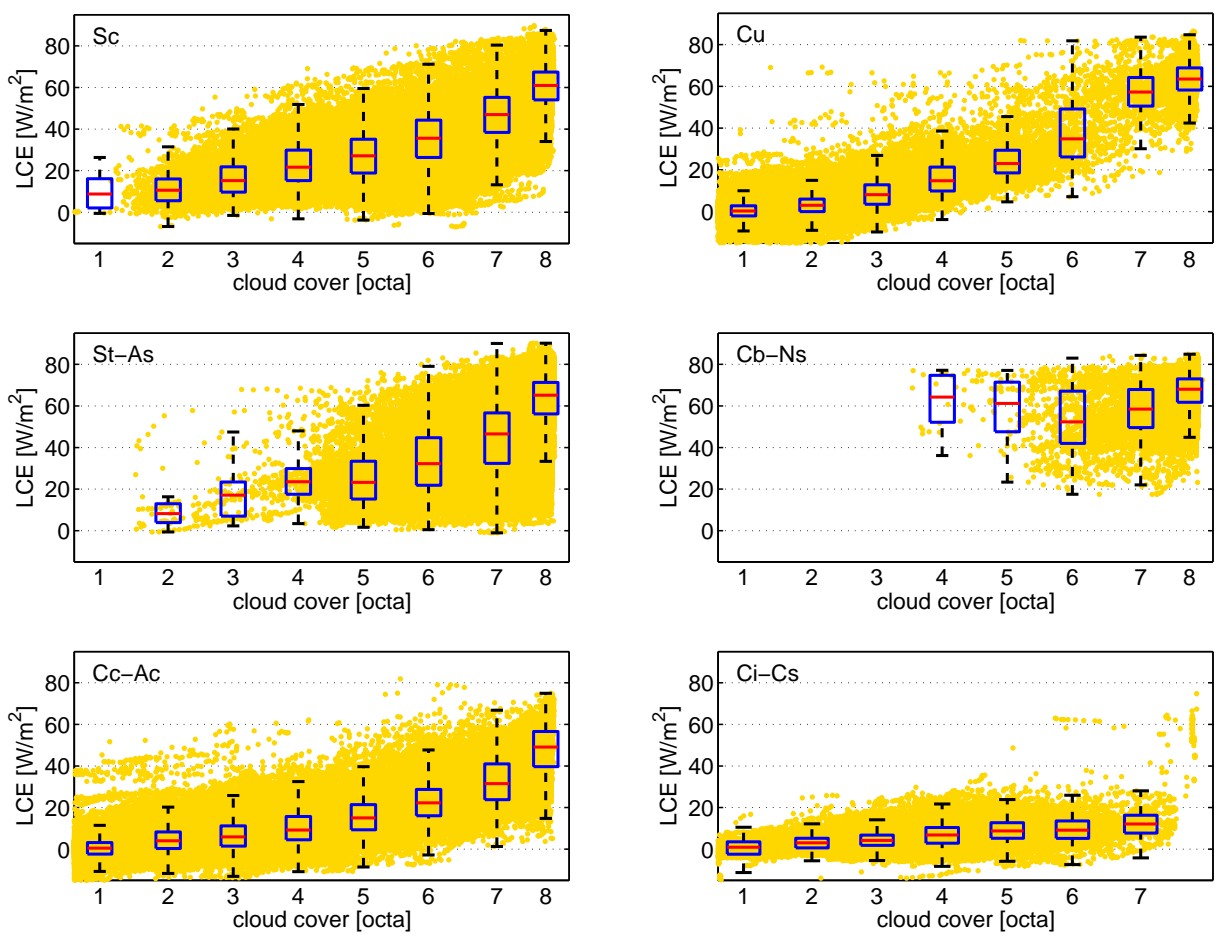

**Figure 3.** Dependence of LCE on cloud coverage for Davos for cloud classes stratocumulus (Sc), cumulus (Cu), stratus-altostratus (St-As), cumulonimbus-nimbostratus (Cb-Ns), cirrocumulus-altocumulus (Cc-Ac) and cirrus-cirrostratus (Ci-Cs). Data points (yellow dots) and box plots per octa with median (red line), interquartile range (blue box) and spread without outliers.





**Table 1.** Median and interquartile range of longwave cloud radiative effect values $[\mathrm{Wm}^{-2}]$ per octa for the two stations Davos (DAV) and Payerne (PAY) and six cloud classes stratocumulus (Sc), cumulus (Cu), stratus-altostratus (St-As), cumulonimbus-nimbostratus (Cb-Ns), cirrocumulus-altocumulus (Cc-Ac) and cirrus-cirrostratus (Ci-Cs).

| cc [octa] | station | Sc $[\mathrm{Wm}^{-2}]$ | Cu $[\mathrm{Wm}^{-2}]$ | St-As $[\mathrm{Wm}^{-2}]$ | Cb-Ns $[\mathrm{Wm}^{-2}]$ | Cc-Ac $[\mathrm{Wm}^{-2}]$ | Ci-Cs $[\mathrm{Wm}^{-2}]$ |
|---|---|---|---|---|---|---|---|
| 1 | DAV | 9 (2,16) | 0 (-2,3) | - (-,-) | - (-,-) | 0 (-2,3) | 1 (-2,4) |
| | PAY | 1 (-3,7) | 1 (-4,6) | 6 (2,8) | - (-,-) | 0 (-4,5) | 0 (-4,4) |
| 2 | DAV | 11 (6,16) | 3 (0,6) | 8 (4,13) | - (-,-) | 4 (0,8) | 3 (1,5) |
| | PAY | 5 (-1,12) | 10 (4,18) | 7 (3,14) | 8 (-1,42) | 5 (0,13) | 4 (0,10) |
| 3 | DAV | 15 (10,22) | 8 (4,13) | 17 (7,23) | - (-,-) | 6 (1,11) | 4 (2,7) |
| | PAY | 4 (-1,14) | 18 (11,26) | 27 (21,35) | 20 (4,38) | 12 (4,22) | 7 (3,13) |
| 4 | DAV | 22 (15,30) | 15 (10,21) | 24 (18,30) | 64 (52,75) | 9 (5,16) | 7 (3,10) |
| | PAY | 19 (7,35) | 22 (16,29) | 37 (28,45) | 33 (11,53) | 18 (9,29) | 10 (5,15) |
| 5 | DAV | 27 (19,35) | 23 (19,29) | 23 (15,33) | 61 (48,71) | 15 (9,21) | 9 (5,13) |
| | PAY | 31 (18,43) | 25 (18,32) | 36 (29,51) | 39 (20,55) | 23 (13,33) | 12 (7,17) |
| 6 | DAV | 36 (26,44) | 35 (26,49) | 32 (22,45) | 52 (42,67) | 22 (16,29) | 9 (5,14) |
| | PAY | 37 (26,49) | 32 (25,40) | 44 (30,68) | 45 (25,57) | 27 (17,38) | 15 (10,22) |
| 7 | DAV | 47 (38,55) | 57 (51,64) | 47 (32,57) | 58 (50,68) | 32 (24,41) | 12 (8,16) |
| | PAY | 44 (32,54) | 57 (37,65) | 65 (54,71) | 51 (37,61) | 33 (23,40) | 17 (12,24) |
| 8 | DAV | 61 (54,67) | 64 (58,69) | 65 (56,71) | 68 (62,73) | 49 (40,57) | - (-,-) |
| | PAY | 57 (47,65) | 59 (52,65) | 71 (66,75) | 61 (52,69) | 36 (23,49) | 20 (15,25) |

cloud classes for Davos and Payerne separately.

In Davos, the highest median LCE for a cloud coverage of 8 octas is observed for the low-level cloud classes Cb-Ns, St-As, Cu and Sc with a maximum influence on the downward longwave radiation at the surface for Cb-Ns (68 $\mathrm{Wm}^{-2}$). The mid-level and thinner cloud class Cc-Ac has a lower median LCE of 49 $\mathrm{Wm}^{-2}$ for a cloud coverage of 8 octas. Clearly lower is the

5 median LCE value for the high-level cloud class Ci-Cs and 7 octas cloud coverage (12 $\mathrm{Wm}^{-2}$). Also for other cloud coverages median LCE values of the three low-level cloud types Sc, Cu and St-As stay in the same range.

Although the numbers differ between the two stations, the same pattern holds also for Payerne, namely that the lower the cloud, the higher the LCE value. Thus for Payerne, the four low-level cloud types (Sc, Cu, St-As and Cb-Ns) and eight octas cloud coverages have median LCE values of 57 - 71 $\mathrm{Wm}^{-2}$ (with interquartile ranges of maximum $\pm 9$ $\mathrm{Wm}^{-2}$). The median

LCE value for the mid-level cloud class Cc-Ac and eight octas cloud coverage is with 36 $\mathrm{Wm}^{-2}$ clearly lower than the values for the low-level clouds and also in comparison with the same values in Davos. The median LCE value for the high-level cloud class Ci-Cs and 8 octas is around 20 $\mathrm{Wm}^{-2}$. This value is only slightly lower for smaller cloud coverages.

The median LCE values of the two stations differ the more the smaller the cloud coverage is. Except Sc and Cb-Ns, the LCE values are generally larger for the station Payerne in comparison with Davos. One part of the difference might be explained

with the fact that the LW clear sky model for Payerne is underestimating more the measurements. Another explanation for this difference might be that Payerne is located at a lower altitude level and thus the cloud base temperature is larger, which leads





**Table 2.** Median and interquartile range of relative shortwave cloud radiative effect values [%] per octa for the two stations Davos (DAV) and Payerne (PAY) and six cloud classes stratocumulus (Sc), cumulus (Cu), stratus-altostratus (St-As), cumulonimbus-nimbostratus (Cb-Ns), cirrocumulus-altocumulus (Cc-Ac) and cirrus-cirrostratus (Ci-Cs).

| cc [octa] | station | Sc [%] | Cu [%] | St-As [%] | Cb-Ns [%] | Cc-Ac [%] | Ci-Cs [%] |
|---|---|---|---|---|---|---|---|
| 1 | DAV | 4 (0,5) | 1 (-2,4) | - (-,-) | - (-,-) | 1 (-1,3) | 1 (-2,4) |
| | PAY | -9 (-32,6) | 1 (-31,9) | -70 (-70,-70) | - (-,-) | 3 (-18,10) | 1 (-9,7) |
| 2 | DAV | 2 (-22,11) | 2 (-8,7) | 11 (7,16) | - (-,-) | 3 (-6,7) | 1 (-3,5) |
| | PAY | -5 (-36,10) | -14 (-53,12) | -43 (-58,-31) | -35 (-69,-4) | -17 (-50,12) | -2 (-17,7) |
| 3 | DAV | -4 (-50,13) | 3 (-31,9) | 16 (7,29) | - (-,-) | 3 (-20,9) | -1 (-7,5) |
| | PAY | -54 (-68,-37) | -27 (-56,12) | -38 (-51,-24) | -57 (-83,-38) | -29 (-51,6) | -9 (-22,6) |
| 4 | DAV | -15 (-51,14) | -12 (-55,12) | 16 (-53,31) | -73 (-79,-70) | -3 (-46,11) | -5 (-18,5) |
| | PAY | -59 (-67,-51) | -42 (-60,4) | -44 (-53,-27) | -46 (-57,-33) | -29 (-48,1) | -10 (-25,3) |
| 5 | DAV | -27 (-53,12) | -46 (-64,-5) | -28 (-51,1) | -70 (-79,-52) | -21 (-54,10) | -6 (-20,4) |
| | PAY | -56 (-64,-45) | -51 (-62,-22) | -34 (-52,-23) | -54 (-75,-37) | -28 (-45,-1) | -12 (-27,0) |
| 6 | DAV | -37 (-55,-6) | -60 (-70,-48) | -40 (-54,-13) | -66 (-76,-46) | -21 (-51,8) | -7 (-18,3) |
| | PAY | -50 (-60,-38) | -44 (-58,-7) | -37 (-60,-17) | -60 (-76,-39) | -24 (-41,1) | -20 (-34,-8) |
| 7 | DAV | -45 (-58,-26) | -70 (-77,-59) | -45 (-57,-27) | -67 (-77,-54) | -37 (-55,-8) | -9 (-18,0) |
| | PAY | -48 (-58,-35) | -60 (-67,-29) | -59 (-71,-43) | -64 (-78,-42) | -25 (-39,0) | -20 (-33,-8) |
| 8 | DAV | -61 (-72,-49) | -77 (-84,-69) | -62 (-75,-48) | -88 (-94,-80) | -66 (-77,-54) | - (-,-) |
| | PAY | -64 (-76,-51) | -67 (-78,-58) | -73 (-80,-66) | -82 (-89,-72) | -48 (-62,-30) | -28 (-41,-16) |

to a larger emission of LW radiation.

### 3.1.2 Shortwave Cloud Effect

The relative shortwave cloud radiative effect ($SCE_{rel}$) is calculated using Eq. 2.

Table 3 summarizes the median of the $SCE_{rel}$ and the corresponding interquartile range for cloud coverages of one to eight octas and for the cloud classes for the two stations Davos and Payerne separately.

In Davos, the cloud type Cb-Ns, with -88 %, is the cloud type with the largest attenuation $SCE_{rel}$ value for eight octas cloud coverage. The second lowest $SCE_{rel}$ value for eight octas cloud coverage is observed for the cloud type Cu (-77 %), followed by Cc-Ac (-66 %). The cloud classes St-As (-62 %) and Sc (-61 %) are almost in the same range. The uncertainty ranges given as interquartile range are for a fully covered sky up to ±14 %. Also here no statistical values have been calculated for the high-level cloud class Ci-Cs and a cloud coverage of 8 octas due to the same explanation as given in Section 3.1.1. However the median SCE for Ci-Cs and 1 to 7 octas cloud coverage is in comparison to the low-level cloud classes clearly higher with values between 0 and -9 %. In general, the median $SCE_{rel}$ values become higher the smaller the cloud coverage is. This behaviour is obtained for all cloud classes except Cb-Ns, where the median $SCE_{rel}$ values stay in the same range for all investigated cloud coverages.

In Payerne, a different order is observed in the lowest to the highest $SCE_{rel}$ values for a cloud coverage of eight octas. The





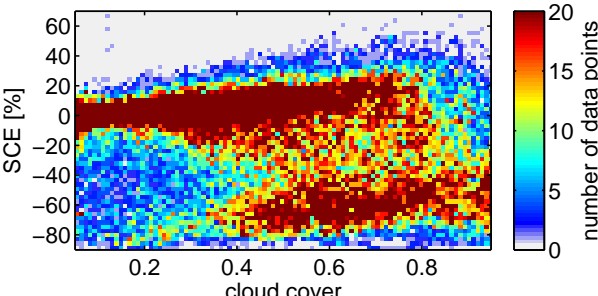

**Figure 4.** Density distribution of the dependence of $SCE_{rel}$ on cloud coverage for Davos for mid-level clouds (Cc-Ac). The density colour distribution represents the number of data points.

cloud class with the lowest values, and thus the larges effect on SW radiation, is again Cb-Ns with -82 %, followed by St-As (-73 %), Cu (-67 %) and Sc (-64 %). The interquartile ranges are in a similar range as the ones for Davos. All these four cloud classes are low-level cloud types and also thicker clouds than the ones at a higher level. Therefore it is reasonable to infer that these are the four cloud classes with the greatest effect on the downward shortwave radiation. For Payerne, a clearly lower

median $SCE_{rel}$ is observed for the mid-level cloud class Cc-Ac and a cloud coverage of eight octas (-48 %) in comparison to low-level clouds. The highest median $SCE_{rel}$ value for 8 octas cloud coverage is observed for the high-level cloud class Ci-Cs (-28 %).

The differences in $SCE_{rel}$ values between Davos and Payerne are for several cloud types and cloud coverages rather high (e.g. 32 % for Cc-Ac and 3 octas). One explanation for these larger differences, mainly for smaller cloud coverages, might be the

so-called cloud enhancement phenomenon, since the positive $SCE_{rel}$ values might increase the median of $SCE_{rel}$. A cloud enhancement phenomenon describes an event where more downward shortwave radiation is measured at the surface under cloudy condition than expected under clear-sky condition. Multiple scattering at cloud particles lead to an increase of the diffuse irradiance part of the shortwave radiation.

Figure 4 shows a density plot of the dependence of $SCE_{rel}$ on fractional cloud coverage in Davos for the mid-level cloud class

Cc-Ac. Mainly with larger cloud coverages there is a region of higher densities of data points of $SCE_{rel}$ values between -80 to -60 %. However, there is another stronger local maximum in the density distribution which shows positive $SCE_{rel}$ values of up to 20 % at smaller cloud coverages. There are also some cases where the $SCE_{rel}$ values reach up to 40 %. This enhancement of the downward shortwave radiation measured at the surface in the presence of clouds can also be detected in the low-level cloud classes.

   If we define a cloud radiative enhancement with a $SCE_{rel}$ of minimum +5 %, in Davos 74,857 cases of the 576,921 cloud cases are detected as cloud enhancement, thus in 13 % of the analysed cases. The cloud class Cc-Ac, with 32 % of the cases,

contributes most to this large number. The cloud class with the second greatest contribution to cloud enhancement is Cu with 27 % followed by Sc (20 %), St-As (11 %) and Ci-Cs (10 %). The cases of observed cloud enhancement due to the presence of Cb-Ns is negligibly small with 0.2 %. Thus the mid-level cloud class Cc-Ac leads to most of the cases of cloud enhancement.





However, checking for the cloud types that produce SCE values of more than 40 % leads to another order of contribution of different cloud classes.

In Davos, 2,621 cases (0.5 % of the cloud data) are observed with $SCE_{rel}$ values of 40 % and above. Here the contribution of the two low-level cloud classes St-As (47 %) and Sc (37 %) is greater than the contribution of the mid-level cloud class Cc-Ac (12 %). These are also the cloud types that mainly contribute to such high positive SCE values. The contributions of Ci-Cs (2 %), Cu (1 %) and Cb-Ns (0.3 %) are negligibly small.

In Payerne, only in 8 % of the 126,148 cloud cases is a cloud enhancement of more than 5 % $SCE_{rel}$ observed. Also here, most
of the cloud enhancement cases are Cc-Ac with 38 % contribution followed by Ci-Cs with 37 %. Cu only make a contribution of 18 % to the total 9,528 cases of cloud enhancement greater than 5 % $SCE_{rel}$. In 7 % of the cloud enhancement cases in Payerne a Sc cloud is responsible. The number of cloud enhancement cases for the cloud classes Cb-Ns (0.7 %) and St-As (0.2 %) is negligibly small.

A cloud enhancement of at least 40 % $SCE_{rel}$ in Payerne is detected only for 412 cases in total in the studied time period. More
than half of these 412 cases are Cc-Ac (58 %), followed by Cu (18 %) and Sc (13 %). Only a few cases are Ci-Cs (7 %) and Cb-Ns (3 %). For St-As clouds there is no case observed with a cloud enhancement of more than 40 % $SCE_{rel}$.

*Schade et al.* (2007) also showed that altocumulus is the cloud type that produces most of the downward solar cloud enhancement. They demonstrated that altocumulus clouds can be responsible for temporary enhancements of up to 500 $Wm^{-2}$. In our data, in Davos the maximum in cloud enhancement with Cc-Ac is a SCE value of 477 $Wm^{-2}$ and in Payerne of 486 $Wm^{-2}$ under Ci-Cs conditions. *Schade et al.* (2007) show that the largest cloud enhancements can be registered at almost overcast situations. However, our data show a maximum in cloud enhancement cases for a fractional cloud coverage of 1 to 2 octas.

The manual analysis of the cloud camera images with cloud enhancement leads to the result that in most of the cases there is a low solar zenith angle. Additionally, it has been observed that in cloud enhancement cases the sun is either in the vicinity of
the cloud or covered with a thin cloud layer.

### 3.1.3    Total Cloud Effect

The total cloud radiative effect (TCE) is calculated as the sum of the LCE and SCE (Eq. 1). The calculated median TCE values and the corresponding interquartile range for cloud coverages of one to eight octas and the cloud classes for the two stations
Davos and Payerne separately are summarized in Table 4. For the calculation of TCE, the absolute values of SCE are taken into account and Eq. 2 is not applied. The TCE values are mainly given to get an idea whether the SCE or the LCE is the prevailing contributor to the TCE during daytime.

During daytime, the SCE values are the main contribution to the TCE for all cloud classes and cloud coverages of 6 to 8 octas and the two stations Davos and Payerne. For the low-level cloud type Cb-Ns, the TCE values are negative for all octas cloud
coverages. Thus during daytime the SCE is the main contributor to TCE for this cloud class. The smaller the cloud coverage is, the more positive the TCE values are. This behaviour can be seen for all cloud types and both stations. Among other reasons,



**Table 3.** The median and interquartile range of the total cloud radiative effect $[Wm^{-2}]$ per octa for the two stations Davos (DAV) and Payerne (PAY) and the six cloud classes stratocumulus (Sc), cumulus (Cu), stratus-altostratus (St-As), cumulonimbus-nimbostratus (Cb-Ns), cirrocumulus-altocumulus (Cc-Ac) and cirrus-cirrostratus (Ci-Cs).

| cc [octa] | station | Sc [Wm$^{-2}$] | Cu [Wm$^{-2}$] | St-As [Wm$^{-2}$] | Cb-Ns [Wm$^{-2}$] | Cc-Ac [Wm$^{-2}$] | Ci-Cs [Wm$^{-2}$] |
|---|---|---|---|---|---|---|---|
| 1 | DAV | 25 (-2,38) | 5 (-9,20) | - (-,-) | - (-,-) | 4 (-5,15) | 4 (-10,23) |
| | PAY | -32 (-103,37) | 7 (-108,58) | -156 (-156,-156) | - (-,-) | 17 (-64,62) | 6 (-45,36) |
| 2 | DAV | 19 (-76,65) | 12 (-29,42) | 69 (40,84) | - (-,-) | 17 (-23,44) | 6 (-14,30) |
| | PAY | -26 (-159,66) | -46 (-228,90) | -100 (-224,-80) | -83 (-149,-19) | -54 (-155,72) | -10 (-77,42) |
| 3 | DAV | -6 (-186,86) | 21 (-109,68) | 82 (44,121) | - (-,-) | 17 (-72,62) | -1 (-27,25) |
| | PAY | -197 (-278,-112) | -115 (-293,94) | -78 (-142,-51) | -91 (-241,-60) | -86 (-160,39) | -38 (-109,41) |
| 4 | DAV | -40 (-201,93) | -40 (-214,91) | 76 (-83,134) | -58 (-95,-52) | -9 (-166,78) | -15 (-67,28) |
| | PAY | -192 (-268,-134) | -198 (-359,55) | -118 (-223,-58) | -91 (-173,-23) | -89 (-184,26) | -49 (-135,31) |
| 5 | DAV | -73 (-227,83) | -131 (-310,-5) | -64 (-135,24) | -92 (-199,-58) | -79 (-234,81) | -25 (-80,28) |
| | PAY | -188 (-290,-116) | -281 (-415,-119) | -114 (-210,-40) | -116 (-218,-59) | -95 (-194,18) | -65 (-152,9) |
| 6 | DAV | -119 (-287,8) | -187 (-353,-77) | -94 (-169,-18) | -105 (-186,-39) | -89 (-243,80) | -31 (-84,23) |
| | PAY | -170 (-273,-100) | -257 (-364,-15) | -135 (-191,-60) | -138 (-262,-85) | -86 (-179,25) | -120 (-202,-36) |
| 7 | DAV | -200 (-342,-69) | -238 (-435,-124) | -127 (-234,-51) | -134 (-238,-60) | -163 (-301,-25) | -45 (-103,7) |
| | PAY | -169 (-271,-100) | -303 (-401,-101) | -167 (-268,-121) | -161 (-249,-91) | -86 (-200,28) | -118 (-207,-34) |
| 8 | DAV | -323 (-455,-190) | -310 (-504,-185) | -225 (-377,-126) | -210 (-356,-112) | -277 (-436,-162) | - (-) |
| | PAY | -255 (-381,-160) | -466 (-560,-323) | -263 (-399,-171) | -226 (-365,-160) | -225 (-341,-99) | -181 (-271,-95) |

one reason for these positive values with smaller cloud coverages might be the cloud enhancement events as described in section 3.1.2.

## 3.2 Sensitivity Analysis

### 3.2.1 Longwave Cloud Effect

As seen in Figure 3, the spread of the data within one octa cloud coverage is large. This large spread can be explained for example by the misclassification of the cloud type as well as by the uncertainty of the detection of cloud fraction of $\pm 1$ octa (*Wacker et al.*, 2015). Additionally, other parameters are responsible for this uncertainty. Thus in a sensitivity analysis the influence of integrated water vapour (IWV) and cloud base height (CBH) is analysed.

Figure 5 shows the dependence of LCE on changes of IWV for all low-level clouds (Sc, Cu, St-As and Cb-Ns) and a cloud
coverage of eight octas for Davos. The low-level clouds have been taken together since on the one hand the LCE values for all the four low-level cloud classes are in a similar range and on the other hand there is considerable uncertainty in the distinguishing of the different cloud classes with increasing cloud coverage using the sky camera images. Figure 5 shows a slightly negative trend between the LCE and IWV. The higher the water vapour content in the atmosphere, the lower are the values of the LCE. Although the trend is statistically not significant, this negative trend is detected for different cloud classes, fractional
cloud coverages and for the two stations Davos and Payerne.





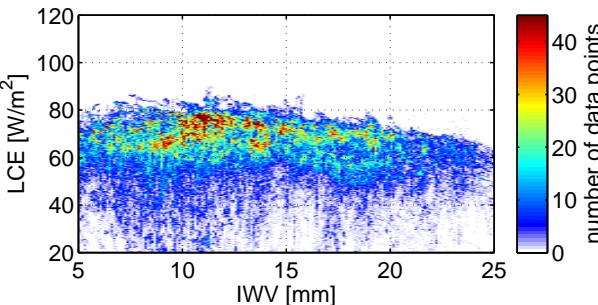

**Figure 5.** Dependence of LCE on integrated water vapour (IWV) for Davos and cloud coverage of 8 octas for low-level clouds (Sc, Cu, St-As, Cb-Ns) shown as a density plot.

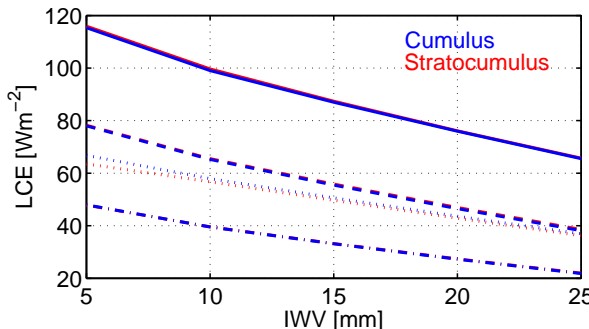

**Figure 6.** Dependence of LCE on integrated water vapour (IWV) modelled for cumulus (blue) and stratocumulus (red) clouds. Solid line: summer standard atmosphere (SSA) and cloud base height (CBH) of 1 km. Dotted line: SSA, CBH = 5 km. Dashed line: winter standard atmosphere (WSA), CBH = 1 km. Dash-dotted line: WSA, CBH = 5 km.

The observed relationship between LCE and IWV was analysed by modelling a standard situation with the moderate resolution atmospheric transmission model MODTRAN5 (*Berk et al.*, 2005). We assume a standard atmosphere profile for mid-latitude summer and winter separately with 50 altitude levels. We also assume no aerosol extinction throughout the atmosphere, due to its negligible influence on the longwave radiation (*Ramanathan et al.*, 2001; *di Sarra et al.*, 2011). The default cloud parameters that have been taken for the model are for cumulus, a cloud thickness of 2.34 km (stratocumulus: 1.34 km), a cloud extinction coefficient at 0.55 $\mu$m of 92.6 km$^{-1}$ (38.7 km$^{-1}$) and a cloud liquid water vertical column density of 1.6640 kg m$^{-2}$ (0.2165 kg m$^{-2}$) respectively. The input IWV values have been changed between 5 and 25 mm. The output of the model is shown in Figure 6 for cumulus (blue) and stratocumulus (red).

The mean values of the observed dependence of LCE on IWV (Figure 5) agree well with the mean values of the modelled dependence of the two aforementioned parameters LCE and IWV (Figure 6). Also the model shows that more water vapour in the atmosphere results in lower LCE values for the two cloud types. The influence is smaller because in cases where there is more water vapour in the atmosphere, the cloud is shielded and the longwave radiation measured at the Earth's surface is





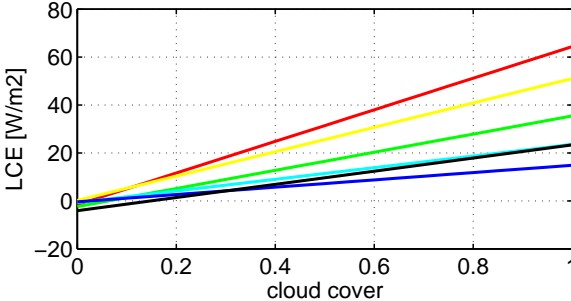

**Figure 7.** Dependence of LCE on cloud coverage for Payerne and linear regression lines of the following cloud base height ranges of lowest measured CBH: red: 0 - 2000 m, yellow: 2000 - 4000 m, green: 4000 - 6000 m, cyan: 6000 - 8000 m, blue: 8000 - 10000 m and black: 10000 m and above.

partially coming from the water vapour and partially from the cloud itself. In the case of less IWV in the atmosphere, the influence of the cloud is greater and consequently also the LCE is higher. Cu and Sc show a similar behaviour in the model which might be explained by the similar shape of the two cloud types.

Another parameter which might explain the large spread in the LCE within one cloud cover range is the cloud base height
10  (CBH). This analysis has only been performed for the data set in Payerne, because it is only at this location that we measure the CBH with a ceilometer. The observed mean dependence of LCE on fractional cloud coverage and CBH is shown in Figure 7. Different colours represent an interval of 2000 m in measured CBH (red: 0 - 2 km, to black: above 12 km).

Figure 7 shows that the lower the CBH, the higher is the LCE. This pattern can be explained by the fact that a lower CBH is a proxy for a higher cloud base temperature which in turn leads to higher thermal emission. The modelling of these cases
15  with the radiative transfer model MODTRAN5 with the same standard conditions as explained in Section 3.2.1 confirms this assumption. The influence of CBH on downward longwave radiation has been analysed in more detail in (*Viudez-Mora et al.*, 2015).

Another important parameter in the LCE discussion for thin clouds is the optical depth of clouds (*Viudez-Mora et al.*, 2015). However, since no data of this parameter are available, it is not discussed in the current study.

### 3.2.2 Shortwave Cloud Effect

In Figure 4 it has been shown that mainly for small cloud coverages the majority of the cases show a $SCE_{rel}$ value of around 0 %. In order to understand these values and the difference in the situation when the $SCE_{rel}$ value is in a strong negative range we analysed the images to determine whether the sun is directly covered by a cloud or not. Whether the sun is covered or
uncovered is decided on the basis of measured data of direct solar irradiance. In cases where the value of the direct solar irradiance measurement of $120\,\mathrm{Wm^{-2}}$ per time step is exceeded, it is assumed that the sun is not covered by a cloud. This reference value of $120\,\mathrm{Wm^{-2}}$ is defined by the World Meteorological Organization (*CIMO*, 2014).



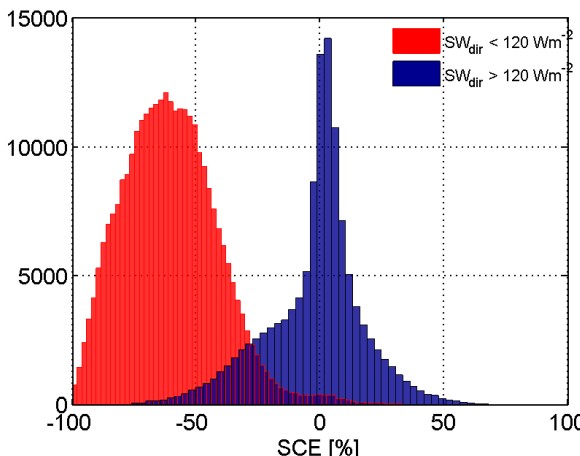

**Figure 8.** Distribution of SCE values for Davos for low-level clouds (Sc, Cu, St-As, Cb-Ns). The measured direct SW radiation is below (red) or above (blue) a threshold of 120 $\mathrm{Wm^{-2}}$.

Figure 8 shows the distribution of $\mathrm{SCE_{rel}}$ values of all data points in Davos for low-level clouds (Sc, Cu, St-As and Cb-Ns). This distribution shows two peaks, one at around $\mathrm{SCE_{rel}}$ values of 0 % and the other one at $\mathrm{SCE_{rel}}$ values of -65 %.

If the cases are now divided into cases where the measured direct radiation value is below 120 $\mathrm{Wm^{-2}}$ (red) and above this threshold (blue), the result is two separate histograms as shown in Figure 8. The red histogram shows the situations in which the cloud has a substantial effect on decreasing the measured shortwave radiation at the surface which results in a more negative $\mathrm{SCE_{rel}}$ value. The peak from the blue histogram is around zero to slightly positive values, there the sun is uncovered and thus the cloud is not diminishing the direct radiation but rather increasing the diffuse radiation measured at the surface.

## 4   Conclusions and Outlook

The current study describes a time series of the cloud radiative effect depending on cloud type and cloud fraction at two stations in Switzerland. Furthermore, it explains the influence of integrated water vapour, the cloud base height and the coverage of the sun on the cloud radiative effect.

Clouds increase the downward longwave radiation measured at the surface of the Earth and in general, decrease the downward shortwave radiation. Different cloud types have differing effects on the radiation measured on the Earth's surface. We have shown that low-level cloud types like cumulus, stratocumulus, stratus-altostratus and cumulonimbus-nimbostratus have with median values of 57 - 71 $\mathrm{Wm^{-2}}$ greater longwave cloud radiative effect values than for example mid-level clouds cirrocumulus-altocumulus (37 - 49 $\mathrm{Wm^{-2}}$). Our measurements show that most low-level cloud types have a longwave cloud effect at the surface in a similar range. The differences in the longwave cloud radiative effect between the two stations Davos and Payerne is



for a cloud coverage of 8 octas up to 7 $\mathrm{Wm}^{-2}$ and is becoming even larger (up to around 25 $\mathrm{Wm}^{-2}$) the smaller the fractional cloud coverage is. It has been shown that the cloud base height and the fractional cloud coverage have an influence on the range of the LCE. The higher the cloud coverage, the greater the LCE and the lower the cloud base height, the larger the LCE. The cloud base height can be taken as a proxy for cloud temperature and thermal emissivity.

However, not only cloud parameters like coverage and CBH have an influence, but also other atmospheric parameters. Our measurements show that there is a negative dependence of the LCE on integrated water vapour. A similar trend was observed using radiative transfer modelling studies. A similar trend was also demonstrated by *Wacker et al.* (2011) for stratus nebulosus. We have shown that the LCE of not only low-level clouds but also of the mid-level cloud class cirrocumulus-altocumulus show a dependence on cloud fraction, CBH and IWV.

It has been shown, that the greater the cloud coverage is, the smaller are the differences of LCE values among cloud types. Thus, for a future study it might be enough to distinguish between low-, mid- and high-level clouds instead of cloud type.

Also a time series of the SCE has been produced for the six cloud classes. It has been shown that also for the SCE the parameters cloud coverage and cloud type and thus cloud thickness influence the magnitude of the $SCE_{rel}$. Thus, low-level clouds have a greater effect on the SCE (up to - 88 % for Cb-Ns) than mid- (up to - 66 %) or high-level clouds (- 28 %). However, not only cloud parameters have an influence, but also whether the sun is covered or not by a cloud. Our study shows that there are two different distributions depending on whether the measured direct SW radiation exceeds a threshold of 120 $\mathrm{Wm}^{-2}$ or not. One of the distributions has a peak at around -60 % $SCE_{rel}$ and the uncovered cases have a peak at around 0 % $SCE_{rel}$ for Davos for the cloud type Cc-Ac. This difference is that high because the direct part of the shortwave radiation contributes most to the total shortwave radiation at the surface. Not only the mid-level cloud class shows these two peaks, but also all the low-level cloud classes Cu, Sc, St-As and Cb-Ns individually. For each of these low-level cloud classes the two peaks of the $SCE_{rel}$ values are also in the same range. The differences in median $SCE_{rel}$ values between Davos and Payerne are even larger than for the LCE, with in general higher $SCE_{rel}$ values for Payerne.

Our data show that in 13 % and 8 % of the cases in Davos and Payerne respectively a shortwave cloud radiative enhancement of at least 5 % is observed. We showed that Cc-Ac is the cloud type that is responsible for around one third of the cloud enhancement cases in Davos and Payerne. Several studies (e.g. *Robinson* (1966); *Schade et al.* (2007); *Thuillier et al.* (2013); *Calbo et al.* (2017)) show the influence of the magnitude of cloud enhancement events and its duration. To compare our results with these analyses about the duration of cloud enhancement events the resolution of 1 min images needs to be decreased to the seconds range and will be subject of a subsequent study.

In the current analysis, only one cloud type per cloud camera image is defined. A step forward would be to distinguish between different cloud types per image. This detection of different cloud types per image is already an intermediate step in our algorithm. At the current state the cloud type with most of the hits is determined. A further advance would be to not only get the most probable cloud type per image but also to obtain the different cloud types per image as output. Thereafter a more accurate analysis considering the influence of the cloud type on the cloud radiative effect would be possible.

So far we have only calculated the longwave, the shortwave and the total cloud radiative effect for daytime observations. Since the SCE is the larger effect during daytime, but is zero at nighttime, in order to calculate the total cloud radiative effect and to





20 make the energy budget complete, also data for the nighttime have to be considered. In this direction a new instrument (infrared cloud camera) has been developed in order to collect all-sky cloud information from nighttime measurements.

*Competing interests.* The authors declare that they have no conflict of interest.

*Acknowledgements.* This research was carried out within the framework of the project A Comprehensive Radiation Flux Assessment
25 (CRUX) financed by MeteoSwiss. Alexander Haefele from MeteoSwiss provided the ceilometer data. All data are available from the corresponding author on request.



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
