# Peer review of "Cloud radiative effect, cloud fraction and cloud type at two stations in Switzerland using hemispherical sky cameras"

_Atmospheric Measurement Techniques, 2017_

## Referee Comment (RC1) · J. Calbó (Referee) · 14 Aug 2017

This paper presents a summary of radiation measurements performed at two sites in Switzerland, in combination with estimations of cloud cover and type based upon an automatic method performed on whole sky images. Specifically, radiation measurements are presented as cloud radiative effect, as the corresponding modelled irradiances for a cloudless sky are subtracted from measurements. Although the literature on cloud radiative effect is pretty large, there is still room for more studies that add insight on this matter, specially for observational studies from ground-based measurements. So this study is worth to be published, although a number of issues should be addressed

before publication. I must say that in general I enjoyed reading the manuscript and that all my comments below are provided with the intention of further improving this study.

General comments:

- Cloud radiative effects are computed by subtracting model estimations of cloudless sky shortwave and longwave irradiances from the corresponding measurements. Therefore, the performance of the "cloudless" models is critical to get suitable values of CRE. The authors give the mean bias of models for both sites, but I suggest that more detail about the performance of cloudless sky models is shown in the paper. It should be quite easy, just by showing the CRE computed for cases corresponding to 0 octas. This could be shown as function of SZA (for shortwave) or as function of month or temperature (for longwave). If the models were correct, the CRE for these cases should be 0 (or at least, centered at 0). If there is a systematic bias at either of the sites, for some SZA, etc., this could be used to further discuss the results. You should also clarify if the "clear sky cases" that you use to assess the models are the same that are later defined as "cloud-free" cases.

- Some deeper discussion of results is needed. In particular, there are some important differences between the two sites, and some strange behavior of CRE that should be highlighted and commented. The authors already make some comments, but additional insight would be appreciated. For example, regarding cloud type (figure 2), there are almost no Cu (and very few St-As) at Payerne, while there are almost no Cb-Ns at Davos. Or, Table 2 shows, for St-As class, that while in Davos enhancements (i.e. CRE > 0) are found for cc < 5, in Payerne CRE reaches very low values (CRE < -35%). It is particularly strange the value for cc = 1, as the median is equal to the first and third quartile (-70%). This also affects results in Table 3, where the behavior at Davos and Payerne is strangely different, in particular for cc < 5 and for most cloud type classes. I wonder if this might be the result of a bias in the cloudless irradiance estimation at one of the two sites (see my previous point) or also a consequence of a very limited number of cases for some particular conditions of cc and cloud type. I mean, for statistics to be

somewhat representative, a minimum number of instances should be included; moreover, a number of instances corresponding to different seasons, years, etc., would be convenient.

Specific Comments, minor suggestions, technical corrections

- Title. I don't think the word "long-term" reflects the content of the study, which is performed on 3-4 years of observations. In fact, no further attention is put on the length of the time series, so simply removing "long-term" from the title would be adequate.

- Abstract. OK in general. You could add that CBH is from ceilometer and IWV from GPS measurements. You could simplify the writing when referring to occulted (measured direct radiation less than 120 Wm-2) or visible Sun (direct radiation greater than 120 Wm-2).

- 2.2 SCE_CSM is not a radiative effect, as you correctly state when defining this symbol. Therefore, I wouldn't use SCE_CSM, but something as SW_CSM, to avoid possible confusion.

- 2.3. Clear sky models. If aerosol conditions are used in the SW model, the source of aerosol measurements/data should be explained in section 2.1.

- 2.4. Cloud fraction and cloud type. If I understand correctly, LCE is also part of the algorithm for cloud type recognition. Although this may be good for obtaining good classification results, it is quite strange in the frame of the present study, as in this way, the "dependent" variable to be studied (LCE) takes also part in the definition of one of the "independent" variables (cloud type). In other words, some "circularity" is introduced by using LCE as a feature for cloud type discrimination. This could partly explain why dispersion of LCE values depending on cloud type/cover is much lower than dispersion of SCE values.

- 3.1.1. LW cloud effect. Could you at least speculate a reason for the non-linearity shown in Fig. 3?
- 3.1.2. SW cloud effect. The first sentence could be set between parentheses within the current second sentence. I would recall some times that "higher" means "less negative". In fact, in the third paragraph, where you say "For Payerne, a clearly lower…" I think it should say "higher". In general, the use of relative values is "risky", as for large SZA the SW irradiance may take very low values, so (given the unavoidable uncertainties in both measurements and cloudless estimations) the relative SCE_rel may tend to very large values. I would suggest using a maximum SZA (SZA < 80 deg?) for the cases included in the analyses. Maybe the horizon characteristics of the two sites already limit the range of SZA, but this should be explicitly commented in the text.

- 3.2.1, Figure 5. I wonder if it is necessary to show results for Cu and Sc, as these results are almost undistinguishable. In addition, it doesn't make sense to put a CBH of 5 km for a low cloud; maybe results for 0.5 km up to 2.5 km for only one cloud type would be more adequate. In any case, the similar behavior between Cu and Sc might be the result of similar microphysical characteristics, not similar "shape".

- 3.2.1, Figure 6. I think that the black line corresponds to 10 km and above, not to above 12 km as written in the text. It would be nice adding another panel, where the LCE is shown, for the 8 octas cases (i.e., for cc > 0.95), against CBH, and also distinguishing by ranges of IWV. I would say that this could be a quite interesting plot that would complement current Fig. 5 and 6.

- Figure 7 is very interesting, but it is showing that the median value of SCE_rel for a given cloud clover might not very representative of what is happening, since in fact there are two very different effects (reduction and enhancement) depending on whether the Sun is occulted or not. Although you comment these two effects, it should be mentioned that median values in tables 2 and 3 are to be taken with caution.

- Conclusions. As a general comment, I would suggest shortening a little bit this section, by removing some repetitive statements and non-essential results. In fact, most general statements correspond to well-known facts (e.g., "…cloud base height and

fractional cloud coverage have an influence on the range of the LCE...”). When writing this kind of well-known results, it should be stated that the current study is confirming them. In other words, it should be made clearer what it is really a finding of the current study, and what are expected results and known facts that the study is confirming.

---

## Referee Comment (RC2) · Anonymous Referee #2 · 17 Aug 2017

This study deals with the analysis of the cloud radiative effect in Switzerland using two sky camera systems in Davos and Payerne in conjunction with pyranometers, pyrgeometers and precision filter radiometers. The results provide analytical information about the shortwave, longwave and total cloud effect components, while a sensitivity analysis was performed as well.

The overall analysis is sound and after the following minor revisions it could be published in the AMT journal. First of all there is a confusion with the Tables troughout the paper. On page 7, line 26 the authors present the LCE results, so the correct Table is 1 (and not 2). Subsequently, on page 10, line 5 Table 3 need to be replaced with Table

2 (describes the SCE), while on page 12, line 10 the corresponding Table is 3 and not 4 (there is not even such a Table in the manuscript).

On page 4, line 19 it is recommended to add an abbreviation for the "lookup table" as LUT in brackets (LUT) and then replace all the subsequent identical expressions with the "LUT" (e.g. on page 4, line 22; 24; etc). Finally, in Sections 3.2.1 (page 13, line 5) and 3.2.2 (page 15, line 2) it is preferable to mention and use as reference the corresponding sections instead of figures, unless Figures 3 and 4 describe the entirety of Sections 3.1.1 and 3.1.2.

I strongly believe that the topic of this manuscript is interesting and the whole approach will be valuable for the AMT community, so after the above corrections this paper worth to be published in the AMT journal.

---

## Referee Comment (RC3) · Anonymous Referee #3 · 28 Aug 2017

General comments:

The manuscript presents calculations of the cloud radiative effect for different cloud types and cloudiness at two stations in Switzerland. Cloud cover and cloud type have been determined using hemispherical sky cameras. Sensitivity analysis have been conducted to study the impact of integrated water vapor and cloud base height on the long-wave cloud radiative effect (LCE), and the occultation of the sun by clouds on the short-wave cloud radiative effect (SCE).

Clouds are the principal modulator of the radiation budget but remain the largest uncertainty in the estimates of the Earth's changing energy budget. Therefore, such

studies are highly important and relevant in order to quantify the effects of clouds on the radiation budget and to monitor their long-terms changes. In addition, the study demonstrates the current limitations of such automated cloud observing systems and hence serve as a base for future improvements. Indeed, the lack of cloud observations at the surface is an important cause for the uncertainties related to clouds.

The manuscript is well structured and - with some exceptions - clearly written. The literature has been carefully selected and cited. Graphics and tables are clear and the captions self-explanatory. There are some issues with the used language. In particular the conclusions could be improved. When the focus is sharpened towards a more original, better structured and formulated conclusion, and some additional minor revisions will be included, this work will be a very interesting and valuable contribution to the atmospheric science community and is in my opinion absolutely suited for publication in AMT.

Specific comments:

- Abstract: The abstract should point to the most relevant results. LCE and SCErel for low-level clouds and 8 oktas cloud cover are described but no statement about the corresponding TCE is given which gives a quantitative feeling to the reader regarding the overall impact of clouds on the radiation balance. I propose to include the corresponding numbers, for instance in line 3 "The total radiative effect of low-level clouds at 8 oktas cloud coverage has a median value....The median of the corresponding long-wave cloud effect (LCE) is....For mid- and high-level clouds the TCE and LCE are significantly lower ..."

- I see one main reason - apart from the atmospheric parameters - for the substantial spread in the CRE data, particularly in the LCE (e.g., Fig. 3 and Fig.5): The deficiencies in the cloud type classification algorithm itself such as misclassification, and/or the fact that only one cloud type can be determined even if several different cloud classes occur. It is for instance not reasonable why there are almost no Cu and St-As at Payerne and

no Cb-Ns at Davos (see Fig. 2). In addition, it is unlikely that such low LCE values can occur for low level clouds (e.g., Sc, Cu, Cb-Ns) and high cloudiness (> 6 oktas) as indicated in Fig. 3. Similarly in Fig.5, it is unlikely, that LCE below 50 (40) Wm-2 occur for IWV contents < 15 (20) mm (see Fig 6, model calculations). So, it is very likely that all these data points are potential misclassifications. These issues should be addressed in the respective paragraphs (there is only a short statement on p. 13) and in the conclusions. Finally, could you derive from your dataset/figures/model calculations a rough percentage of misclassified cases?

- Conclusions: The conclusions should be shortened and better structured. The listing of well-known issues and repetitions should be avoided (e.g., "Different cloud types have differing effects on the radiation.." or the two sentences on p. 16/17 lines 25/5 and p.17 line 14 have a similar meaning (in case the first sentence refers to differences between the two stations and the latter to the differences between cloud types, it would be helpful to state the sentences at least in the same paragraph. Otherwise, the reader will be confused). Finally, the repetitive use of words and expressions such as "Our measurements/data show/It has been shown" should be minimized). Generally, only the most important results and their implications should be stated. In addition to the described results, I would also clearly state the deficiencies in the cloud type classification algorithm which lead to the large spread in the data, particularly in the LCE (see my previous comments). In fact, the authors do mention this issue in the conclusions but the paragraph appears somehow isolated. In addition, a statement about the methodology how the cloud type classification could be improved would be useful in the conclusions: Is it possible to improve the current cloud classification algorithms (and if yes how) or would it rather be a new algorithm by combining various observing systems/methods which measure/calculate the relevant parameters described in this manuscript (e.g., ceilometer for cloud base height, sky camera for cloud cover, LCE and SCE (i.e. observations and the corresponding cloud-free calculations of longwave and shortwave radiation), solar radiation data for the determination of the occultation of the sun and IWV)? Could the authors comment on these issues?

Technical corrections: some of the spelling and grammatical errors:

- Use "cloud-free" instead of "clear-sky" throughout the manuscript ("clear-sky" refers to a sky without clouds and a low aerosol load. The latter is not necessarily the case, particularly at a site in the Midlands such as Payerne. In addition, this is a study about the effect of clouds and thus I would use here rather the term "cloud-free" instead of "clear-sky").

- Use "oktas" instead of "octas" throughout the manuscript

- Use "longwave/shortwave wavelength range" instead of "longwave/shortwave wavelength region" throughout the manuscript

- p.2, line 18: "wider" instead of "broader"

- p.2, lines 19-21: You may rephrase this sentence, something like: "However, the temporal resolution of satellite products is limited. From the Meteosat Second Generation (MSG) geostationary satellites, for instance, data....(Werkmeister et al., 2015). Therefore and for the validation of cloud products from satellites, ground-based observing systems such as all-sky cameras are necessary."

- p.2, line 31: replace "their" by "sensitivity".

- p.3, line 10: write f in italic (f/8)

- p.3, line 16: traceable to the respective standard groups of the World Radiation Center (WRC)

- p.4, line 2: Equation (1): Maybe add "...= DSRobs - DSRcal,cf + DLRobs - DLR,cal,cf" to the equation, "where DSRobs and DLRobs and DSRcal,cf and DLRcal,cf are the observed and calculated downwelling shortwave and longwave fluxes for the all-sky and the corresponding cloud-free scenes, respectively." Then you can delete "which are both calculated separately". Do you assume for the cloud-free calculations the same atmospheric conditions (e.g., temperature, IWV content) as they were observed

during the corresponding (all-sky) measurements? It is nowhere clearly stated. You may state this also here.

- p.4, line 7: I would delete "usually" (or replace by "always"). Clouds increase always the observed LW radiation, don't they?

- p.4, line 19: libRadtran

- p.4, line 31: Include a sentence how you remove the distortion in the Image

- p.6, line 13: could you state a possible explanation for the opposing cloud-free/overcast conditions in winter and summer at Payerne and Davos? Similarly, after line 23, insert a new paragraph and describe the differences in cloud type between the two stations, e.g., fewer Cu and St-As at Payerne with respect to Davos but much more Cb-Ns, most likely due to deficiencies in the cloud type algorithm.

- p.7, Fig.2 in the legend: Cb instead of Cn

- p.7, line 1. "visual observations": Do you refer to routinely conducted synoptic cloud observations by trained personal, i.e. human observer?

- p.7, line 14: to some extent also for St-As.

- p.7, lines 15-20: I would state the statistics for Ci-Cs and 8 oktas coverage for Davos, even if it is too high. Concerning the causes for this particular case, I do not believe that the erroneous values are due to the fact that the camera is not sensitive to high-level clouds. It is not reasonable that the camera detects high-level clouds with lower cloud coverage (these values seem to be reasonable) but does not for overcast conditions. Thus, I would rephrase lines 15-20 which are anyways partly difficult to understand, e.g. something like: "The median for overcast (8 oktas cloud coverage) Ci-Cs conditions in Davos is clearly too high at XX Wm-2. Manually checked images indicate a misclassification of numerous cases as Ci-Cs instead of a cloud type with a lower cloud base and/or optically thicker clouds. Alternatively, the classification as Ci-Cs could be correct, but various cloud types occur at the same time including clouds with a lower

cloud base/optically thicker clouds resulting in higher LCE values for Ci-Cs. A possible reason for the misclassification could be that the algorithm is trained with a data set from Payerne." Finally, I would delete lines 20-24.

- p.7, line 26: It is Table 1 (instead of 2). It would be also helpful to include the absolute or relative numbers of occurrences for the individual cloud classes and cloud cover-ages (in the same table or in a separate table). Indeed, some results which are not reasonable as discussed before could be also due to a limited number of occurrences for a particular cloud class and cloud cover.

- p. 9, line 10: "at 36 Wm-2" instead of "with 36 Wm-2"

- p. 9, line 13: I would rephrase this sentence, e.g., " The difference of the median LCE values increases with decreasing cloud coverage." or similar.

- p. 9, line 14/15: I would simply write "The difference might be partly due to a higher underestimation of the calculated LW cloud-free irradiances at Payerne." or similar.

- p. 9, line 16: "higher" instead of "larger".

- p. 10, line 4: No new paragraph. Continue directly with "Table 2 summarizes..." on line 4 (and it is Table 2 not Table 3).

- p. 10, line 7: Delete "SCErel value".

- p. 11, line 4: "higher" instead of "lower".

- p. 11, line 12: 2x "conditions" instead of "condition".

- p. 11, line 13: delete "part of the shortwave radiation".

- p. 11, line "at" instead of "with" and "range" instead of "region".

- p. 11, lines 4-6: I would rephrase these two sentences, e.g.: "The largest contribution stems from the cloud class Cc-ac at 32 % of the cases, followed by Cu at 27 %, Sc (20 %), St-As (11 %)....."

- p.11, line 7: "negligibly small at 0.2 %".

- p.12, line 14: "... in 8 % of the 126,148 cloud cases, a cloud enhancement of more than 5 % SCErel is observed."

- p.12, line 25: "Schade et al. (2007) showed..."

- p.12, line 10: "...Davos and Payerne are summarized in Table 3 separately." (it is Table 3).

- p.12, line 12: "...the less negative/the more positive the TCE...".

- p. 12, line 16/p. 13, line 2: "Among other reasons": You may list two or three of them. In addition to the cloud enhancement, the positive values are most likely also due to the relatively large uncertainties in the cloud-free model. In my opinion, this should be stated here.

- p. 17, line 11: "increased" instead of "decreased".

- p. 17/18 lines 18-21: I would rephrase this last paragraph (note: the radiation (not energy) budget would be complete if upwelling fluxes were considered) , something like: "The calculations and observations in this study are limited to daylight hours since the hemispherical sky camera operates in the visible wavelength range. However, for climate-monitoring applications cloud observations during day and night are necessary. Therefore, a new observing system (infrared cloud camera) has been developed..."

---

## Author Comment (AC1) · 9 Oct 2017

We would like to thank the referee for the constructive comments that we have tried to implement in the text. The detailed answers can be find in the attached file.

Please also note the supplement to this comment: https://www.atmos-meas-tech-discuss.net/amt-2017-184/amt-2017-184-AC1-supplement.pdf
* * *

---

## Author Response (AR1)

Dear editor

In the following you will find all our answers to the referee comments. The marked-up manuscript is attached at the end of this file.

We believe that the suggestions and comments of the Referees have substantially helped to improve the paper and we hope that the manuscript, which required minor revision, is acceptable for publication in *Atmospheric Measurement Techniques*.

Kind regards

Christine Aebi

**Reply to comments by J. Calbó (Referee #1)**

on the manuscript " Long-term study of cloud radiative effect, cloud fraction and cloud type at two stations in Switzerland using hemispherical sky cameras " by Aebi et al., submitted to *Atmospheric Measurement Techniques.*

We thank the referee for the constructive comments that we have tried to accommodate in the text. Detailed answers to the comments are given below (bold: referee comment, regular font: author's response).

This paper presents a summary of radiation measurements performed at two sites in Switzerland, in combination with estimations of cloud cover and type based upon an automatic method performed on whole sky images. Specifically, radiation measurements are presented as cloud radiative effect, as the corresponding modelled irradiances for a cloudless sky are subtracted from measurements. Although the literature on cloud radiative effect is pretty large, there is still room for more studies that add insight on this matter, specially for observational studies from ground-based measurements. So this study is worth to be published, although a number of issues should be addressed before publication. I must say that in general I enjoyed reading the manuscript and that all my comments below are provided with the intention of further improving this study.

General comments:

(1) **Cloud radiative effects are computed by subtracting model estimations of cloudless sky shortwave and longwave irradiances from the corresponding measurements. Therefore, the performance of the "cloudless" models is critical to get suitable values of CRE. The authors give the mean bias of models for both sites, but I suggest that more detail about the performance of cloudless sky models is shown in the paper. It should be quite easy, just by showing the CRE computed for cases corresponding to 0 octas. This could be shown as function of SZA (for shortwave) or as function of month or temperature (for longwave). If the models were correct, the CRE for these cases should be 0 (or at least, centered at 0). If there is a systematic bias at either of the sites, for some SZA, etc., this could be used to further discuss the results. You should also clarify if the "clear sky cases" that you use to assess the models are the same that are later defined as "cloud-free" cases.**

Thanks for this comment. The Figure here show in different panels the mean SCE_rel depending on SZA (left) and the mean LCE depending on the screen-level temperature (right) for the cloud-free cases in Davos (top row) and Payerne (bottom row). For the shortwave we see that there is a slight increase in the uncertainty with higher SZA.

We decided to not include this Figure in the paper. However for the shortwave we calculated the mean and the standard deviation separately for SZA < 50° and SZA > 50° and added these values in the text:

p. 4, l. 31 and p.5, l. 1-4:

The difference between SW measurement and the cloud-free model depends on the SZA. The bigger the SZA, the higher the mean difference. In Davos, the mean difference changes

from 7.2 ±20.7 Wm$^{-2}$ (0.9 ±2.6 %) for data with SZA < 50° to 5.7 ±14.7Wm$^{-2}$ (1.1 ±3.8 %) for data with SZA > 50°. In Payerne, the mean difference is 7.3 ±41.7 Wm$^{-2}$ (1.0 ±5.2 %) for data with SZA < 50°. The mean difference is with 3.3 ±34.1Wm$^{-2}$ (0.6 ±8.9 %) slightly larger for data with SZA from 50 to 78°.

Thanks also for the comment about the clear-sky and cloud-free cases. We changed now everything to cloud-free.

[Figure]

(2) Some deeper discussion of results is needed. In particular, there are some important differences between the two sites, and some strange behavior of CRE that should be highlighted and commented. The authors already make some comments, but additional insight would be appreciated. For example, regarding cloud type (figure 2), there are almost no Cu (and very few St-As) at Payerne, while there are almost no Cb-Ns at Davos. Or, Table 2 shows, for St-As class, that while in Davos enhancements (i.e. CRE > 0) are found for cc < 5, in Payerne CRE reaches very low values (CRE < -35%). It is particularly strange the value for cc = 1, as the median is equal to the first and third quartile (-70%). This also affects results in Table 3, where the behavior at Davos and Payerne is strangely different, in particular for cc < 5 and for most cloud type classes. I wonder if this might be the result of a bias in the cloudless irradiance estimation at one of the two sites (see my previous point) or also a consequence of a very limited number of cases for some particular conditions of cc and cloud type. I mean, for statistics to be somewhat representative, a minimum number of instances should be included; moreover, a number of instances corresponding to different seasons, years, etc., would be convenient.

We added some more paragraphs to discuss the different behavior in the data following your suggestions:

p. 7, l. 6ff.:

In Davos, as determined by our algorithm, from October to May St-As is 5 present in at least 40 % of the cases per month. This fraction of St-As is rather too high and might be due to a limitation of the cloud type algorithm. The limitation is, that the algorithm applied for Davos is trained with images from Payerne. Therefore it might be more difficult to distinguish between low-level cloud classes (e.g. St-As and Sc) in Davos. This limitation might also be responsible for the rather infrequent determination of Cu in Davos.

p. 12, l. 5ff.:

For the calculation of the values in Table 2 different numbers of cases have been taken into account (see Table A1 and A2). Analysing e.g. the images that belong to the group St-As and 2 oktas in more detail, leads to the result that at all the images for this specific group in Payerne the sun is covered by a cloud, whereas in Davos, of the 58 images only in around 20 % of the cases the sun is occulted and in the remaining 80 % the sun is visible. As further discussed in Section 3.3.2, this fact of visible or occulted sun can lead to a large difference in $SCE_{rel}$ values. These larger differences in $SCE_{rel}$ values between the two stations mainly occur when only a limited number of images is available. Therefore, some of the $SCE_{rel}$ values have to be taken with caution.

Additionally we added in the appendix two tables (A1 and A2) which show the number of cases that have been taken into account for the calculation of Table 1, Table 2 and Table 3.

**Specific comments:**

(1) **Title. I don't think the word "long-term" reflects the content of the study, which is performed on 3-4 years of observations. In fact, no further attention is put on the length of the time series, so simply removing "long-term" from the title would be adequate.**

We changed the title to:

Cloud radiative effect, cloud fraction and cloud type at two stations in Switzerland using hemispherical sky cameras.

(2) **Abstract. OK in general. You could add that CBH is from ceilometer and IWV from GPS measurements. You could simplify the writing when referring to occulted (measured direct radiation less than 120 Wm$^{-2}$) or visible Sun (direct radiation greater than 120 Wm$^{-2}$).**

The following sentence has been added in the abstract:

p. 1, l. 3-4:

The cloud base height (CBH) information are retrieved from a ceilometer and integrated water vapour (IWV) data from GPS measurements.

As suggested, the two terms occulted and visible sun have been added and are used in the following:

p. 1, l. 9-11:

In cases where the measured direct radiation value is below the threshold of 120 Wm$^{-2}$ (occulted sun) the SCE$_{rel}$ decreases substantially, while cases where the measured direct radiation value is larger than 120 Wm$^{-2}$ (visible sun) lead to a SCE$_{rel}$ of around 0 %.

(3) **2.2 SCE_CSM is not a radiative effect, as you correctly state when defining this symbol. Therefore, I wouldn't use SCE_CSM, but something as SW_CSM, to avoid possible confusion.**

We changed the symbol SCE_CSM to DSR_cfm in Equation 2 and thereafter also in the text:

p. 4, l. 11:
SCE$_{rel}$ = SCE/DSR$_{cfm}$ * 100%

(4) **2.3. Clear sky models. If aerosol conditions are used in the SW model, the source of aerosol measurements/data should be explained in section 2.1.**

In Section 2.1, p. 3, l. 20-21, we already have the sentence:
Aerosol optical depth data are retrieved from precision filter radiometers (PFR, manufactured by PMOD/WRC).

However, to be more clear, we added:

p. 3, l. 20-21:

Aerosol optical depth (AOD) data, used for the shortwave cloud-free model, are retrieved from precision filter radiometers (PFR, manufactured by PMOD/WRC).

(5) **2.4. Cloud fraction and cloud type. If I understand correctly, LCE is also part of the algorithm for cloud type recognition. Although this may be good for obtaining good classification results, it is quite strange in the frame of the present study, as in this way, the "dependent" variable to be studied (LCE) takes also part in the definition of one of the "independent" variables (cloud type). In other words, some "circularity" is introduced by using LCE as a feature for cloud type discrimination. This could partly explain why dispersion of LCE values depending on cloud type/cover is much lower than dispersion of SCE values.**

LCE as a feature in the cloud type detection algorithm has been added in order to better distinguish between low and high level clouds in cases the sky is fully covered. To add the LCE in the algorithm is an advantage, because in fully covered sky images, the textural features do not give enough specific information to distinguish between different cloud levels and therefore cloud types. Thus it would be too difficult to distinguish between different cloud types which would result in many misclassified images. Since LCE values are quite distinct for low and high level clouds, it helps to distinguish the different cloud levels.

(6) **3.1.1. LW cloud effect. Could you at least speculate a reason for the non-linearity shown in Fig. 3?**

We added the sentences:

p. 8f, l. 11ff:

Clouds at different zenith angles in the sky have a stronger or weaker impact on the downward longwave radiation measured at the surface. In case the zenith angles of the clouds are not equally distributed in our analysed time period, this might be a reason for this nonlinearity in LCE. However, we have not analysed it in more detail yet and is subject of a future study.

(7) **3.1.2. SW cloud effect. The first sentence could be set between parentheses within the current second sentence. I would recall some times that "higher" means "less negative". In fact, in the third paragraph, where you say "For Payerne, a clearly lower…" I think it should say "higher". In general, the use of relative values is "risky", as for large SZA the SW irradiance may take very low values, so (given the unavoidable uncertainties in both measurements and cloudless estimations) the relative SCE_rel may tend to very large values. I would suggest using a maximum SZA (SZA < 80 deg?) for the cases included in the analyses. Maybe the horizon characteristics of the two sites already limit the range of SZA, but this should be explicitly commented in the text.**

We tried to make it more clear that for the SCE_rel higher means less negative.

We decreased the maximum SZA for Davos and we specified the SZA ranges taken in section 2.1, p. 3, l. 25ff.:

Data have only been taken into account for daytime measurements when the sun is located minimum five degrees above the horizon and the mountains. For Payerne, the study of CRE includes data from January 1, 2013 to April 30, 2017 with a time resolution of five minutes. Data considered are during daytime with a solar zenith angle (SZA) of maximum 78°.

(8) **3.2.1, Figure 5. I wonder if it is necessary to show results for Cu and Sc, as these results are almost undistinguishable. In addition, it doesn't make sense to put a CBH of 5 km for a low cloud; maybe results for 0.5 km up to 2.5 km for only one cloud type would be more adequate. In any case, the similar behavior between Cu and Sc might be the result of similar microphysical characteristics, not similar "shape".**

With Figure 6 (we guess that you meant this one), we want to show the influence of IWV on the LCE in general and not for a specific case or station. Thus, we think that this graph shows nicely this influence.

We changed the sentence as suggested (p.16, l. 10-11):

Cu and Sc show a similar behaviour in the model which might be explained by similar microphysical characteristics of the two cloud types.

(9) **3.2.1, Figure 6. I think that the black line corresponds to 10 km and above, not to above 12 km as written in the text. It would be nice adding another panel, where the LCE is shown, for the 8 octas cases (i.e., for cc > 0.95), against CBH, and also distinguishing by ranges of IWV. I would say that this could be a quite interesting plot that would complement current Fig. 5 and 6.**

We decided to remove the current Figure 7 and put a new one which shows the dependence of LCE on cloud base height for Payerne and linear regression lines of the following measured IWV ranges.

(10)      Figure 7 is very interesting, but it is showing that the median value of SCE_rel for a given cloud clover might not very representative of what is happening, since in fact there are two very different effects (reduction and enhancement) depending on whether the Sun is occulted or not. Although you comment these two effects, it should be mentioned that median values in tables 2 and 3 are to be taken with caution.

We mentioned now that the median values in Table 2 have to be taken with caution (p. 12, l. 10-11). The reason why we still think that it makes sense to calculate the median with all data (reduction and enhancement) is, that e.g. in weather prediction models the input about clouds is an average over a certain time period where also enhancements and reductions occur.

(11)      Conclusions. As a general comment, I would suggest shortening a little bit this section, by removing some repetitive statements and non-essential results. In fact, most general statements correspond to well-known facts (e.g., "…cloud base height and fractional cloud coverage have an influence on the range of the LCE…"). When writing this kind of well-known results, it should be stated that the current study is confirming them. In other words, it should be made clearer what it is really a finding of the current study, and what are expected results and known facts that the study is confirming.

We have shortened and changed a large fraction of the conclusion.

**Reply to comments by Anonymous Referee #2**

on the manuscript " Long-term study of cloud radiative effect, cloud fraction and cloud type at two stations in Switzerland using hemispherical sky cameras " by Aebi et al., submitted to *Atmospheric Measurement Techniques*.

We thank the referee for the constructive comments that we have tried to accommodate in the text. Detailed answers to the comments are given below (bold: referee comment, regular font: author's response).

This study deals with the analysis of the cloud radiative effect in Switzerland using two sky camera systems in Davos and Payerne in conjunction with pyranometers, pyrgeometers and precision filter radiometers. The results provide analytical information about the shortwave, longwave and total cloud effect components, while a sensitivity analysis was performed as well.
The overall analysis is sound and after the following minor revisions it could be published in the AMT journal.

(1) **First of all there is a confusion with the Tables throughout the paper. On page 7, line 26 the authors present the LCE results, so the correct Table is 1 (and not 2). Subsequently, on page 10, line 5 Table 3 need to be replaced with Table 2 (describes the SCE), while on page 12, line 10 the corresponding Table is 3 and not 4 (there is not even such a Table in the manuscript).**

We acknowledge your comment on that, it was a compilation error and the table numbers are in the right order now.

(2) **On page 4, line 19 it is recommended to add an abbreviation for the "lookup table" as LUT in brackets (LUT) and then replace all the subsequent identical expressions with the "LUT" (e.g. on page 4, line 22; 24; etc).**

We added the suggested abbreviation LUT and used it then throughout the whole manuscript.

(3) **Finally, in Sections 3.2.1 (page 13, line 5) and 3.2.2 (page 15, line 2) it is preferable to mention and use as reference the corresponding sections instead of figures, unless Figures 3 and 4 describe the entirety of Sections 3.1.1 and 3.1.2.**

We changed the reference to the corresponding sections (instead of the reference to the figures).

**Reply to comments by Anonymous Referee #3**

on the manuscript " Long-term study of cloud radiative effect, cloud fraction and cloud type at two stations in Switzerland using hemispherical sky cameras " by Aebi et al., submitted to *Atmospheric Measurement Techniques.*

We thank the referee for the constructive comments that we have tried to accommodate in the text. Detailed answers to the comments are given below (bold: referee comment, regular font: author's response).

**General comments:**

(1) **The manuscript presents calculations of the cloud radiative effect for different cloud types and cloudiness at two stations in Switzerland. Cloud cover and cloud type have been determined using hemispherical sky cameras. Sensitivity analysis have been conducted to study the impact of integrated water vapor and cloud base height on the long-wave cloud radiative effect (LCE), and the occultation of the sun by clouds on the short-wave cloud radiative effect (SCE).**
**Clouds are the principal modulator of the radiation budget but remain the largest uncertainty in the estimates of the Earth's changing energy budget. Therefore, such studies are highly important and relevant in order to quantify the effects of clouds on the radiation budget and to monitor their long-terms changes. In addition, the study demonstrates the current limitations of such automated cloud observing systems and hence serve as a base for future improvements. Indeed, the lack of cloud observations at the surface is an important cause for the uncertainties related to clouds.**
**The manuscript is well structured and - with some exceptions - clearly written. The literature has been carefully selected and cited. Graphics and tables are clear and the captions self-explanatory. There are some issues with the used language. In particular the conclusions could be improved. When the focus is sharpened towards a more original, better structured and formulated conclusion, and some additional minor revisions will be included, this work will be a very interesting and valuable contribution to the atmospheric science community and is in my opinion absolutely suited for publication in AMT.**

We appreciate your comments. The conclusion has been shortened and partially rewritten.

**Specific comments:**

(2) **Abstract: The abstract should point to the most relevant results. LCE and SCErel for low-level clouds and 8 oktas cloud cover are described but no statement about the corresponding TCE is given which gives a quantitative feeling to the reader regarding the overall impact of clouds on the radiation balance. I propose to include the corresponding numbers, for instance in line 3 "The total radiative effect of low-level clouds at 8 oktas cloud coverage has a median value....The median of the corresponding long-wave cloud effect (LCE) is....For mid- and high-level clouds the TCE and LCE are significantly lower ..."**

Indeed, the total cloud radiative effect (TCE) has not been included in the abstract so far. Following your suggestion we added the following sentence about the TCE on p. 1, l. 13-14:

The calculated median total cloud radiative effect (TCE) values are negative for almost all cloud classes and cloud coverages.

(3) I see one main reason - apart from the atmospheric parameters - for the substantial spread in the CRE data, particularly in the LCE (e.g., Fig. 3 and Fig.5): The deficiencies in the cloud type classification algorithm itself such as misclassification, and/or the fact that only one cloud type can be determined even if several different cloud classes occur. It is for instance not reasonable why there are almost no Cu and St-As at Payerne and no Cb-Ns at Davos (see Fig. 2). In addition, it is unlikely that such low LCE values can occur for low level clouds (e.g., Sc, Cu, Cb-Ns) and high cloudiness (> 6 oktas) as indicated in Fig. 3. Similarly in Fig.5, it is unlikely, that LCE below 50 (40) Wm-2 occur for IWV contents < 15 (20) mm (see Fig 6, model calculations). So, it is very likely that all these data points are potential misclassifications. These issues should be addressed in the respective paragraphs (there is only a short statement on p. 13) and in the conclusions. Finally, could you derive from your dataset/figures/model calculations a rough percentage of misclassified cases?

The misclassification of images indeed leads to an uncertainty in the results. This problem has been added at several places throughout the whole manuscript (e.g. p. 10, l. 4).

The uncertainty of the cloud type classification algorithm has been given on p. 6 l. 5ff.

(4) Conclusions: The conclusions should be shortened and better structured. The listing of well-known issues and repetitions should be avoided (e.g., "Different cloud types have differing effects on the radiation.." or the two sentences on p. 16/17 lines 25/5 and p.17 line 14 have a similar meaning (in case the first sentence refers to differences between the two stations and the latter to the differences between cloud types, it would be helpful to state the sentences at least in the same paragraph. Otherwise, the reader will be confused). Finally, the repetitive use of words and expressions such as "Our measurements/data show/It has been shown" should be minimized). Generally, only the most important results and their implications should be stated. In addition to the described results, I would also clearly state the deficiencies in the cloud type classification algorithm which lead to the large spread in the data, particularly in the LCE (see my previous comments). In fact, the authors do mention this issue in the conclusions but the paragraph appears somehow isolated. In addition, a statement about the methodology how the cloud type classification could be improved would be useful in the conclusions: Is it possible to improve the current cloud classification algorithms (and if yes how) or would it rather be a new algorithm by combining various observing systems/methods which measure/calculate the relevant parameters described in this manuscript (e.g., ceilometer for cloud base height, sky camera for cloud cover, LCE and SCE (i.e. observations and the corresponding cloud-free calculations of longwave and shortwave radiation), solar radiation data for the determination of the occultation of the sun and IWV)? Could the authors comment on these issues?

We have shortened and rewritten the conclusion and outlook part.

Technical corrections: some of the spelling and grammatical errors:

(1) Use "cloud-free" instead of "clear-sky" throughout the manuscript ("clear-sky" refers to a sky without clouds and a low aerosol load. The latter is not necessarily the case, particularly at a

site in the Midlands such as Payerne. In addition, this is a study about the effect of clouds and thus I would use here rather the term "cloud-free" instead of "clear-sky").

Throughout the manuscript we changed all the terms clear-sky to cloud-free.

(2) Use "oktas" instead of "octas" throughout the manuscript

Done

(3) Use "longwave/shortwave wavelength range" instead of "longwave/shortwave wavelength region" throughout the manuscript

Done

(4) p.2, line 18: "wider" instead of "broader"

Done

(5) p.2, lines 19-21: You may rephrase this sentence, something like: "However, the temporal resolution of satellite products is limited. From the Meteosat Second Generation (MSG) geostationary satellites, for instance, data....(Werkmeister et al., 2015). Therefore and for the validation of cloud products from satellites, ground-based observing systems such as all-sky cameras are necessary."

The two sentences have been changed as suggested (p. 2, l. 21ff.):

However, the temporal resolution of satellite products is limited. From the Meteosat Second Generation (MSG) geostationary satellites, for instance, data about clouds are taken with a time resolution of 15 minutes (Werkmeister, 2015). Therefore and for the validation of cloud products from satellites, ground-based observing systems such as all-sky cameras are necessary.

(6) p.2, line 31: replace "their" by "sensitivity".

Done

(7) p.3, line 10: write f in italic (f/8)

Done

(8) p.3, line 16: traceable to the respective standard groups of the World Radiation Center (WRC)

Done

(9) p.4, line 2: Equation (1): Maybe add "...= DSRobs - DSRcal,cf + DLRobs - DLR,cal,cf" to the equation, "where DSRobs and DLRobs and DSRcal,cf and DLRcal,cf are the observed and calculated downwelling shortwave and longwave fluxes for the all-sky and the corresponding cloud-free scenes, respectively." Then you can delete "which are both calculated separately".

Do you assume for the cloud-free calculations the same atmospheric conditions (e.g., temperature, IWV content) as they were observed during the corresponding (all-sky) measurements? It is nowhere clearly stated. You may state this also here.

As suggested, we added to Eq. 1 the following part:
TCE = SCE + LCE = DSR_obs - DSR_cfm + DLR_obs - DLR_cfm

Yes, we assume the same atmospheric conditions (temperature, IWV, etc.) under cloud-free conditions as under cloudy conditions. Therefore we added the following sentence (p. 4, l. 8ff.):

The atmospheric conditions (namely temperature and IWV) in the models are assumed to be the same under cloudy and cloud-free condition.

(10)     p.4, line 7: I would delete "usually" (or replace by "always"). Clouds increase always the observed LW radiation, don't they?

We changed the sentence to (p. 4, l. 13-14):

Clouds increase the measured LW radiation at the surface as they emit LW radiation.

(11)     p.4, line 19: LibRadtran

Done

(12)     p.4, line 31: Include a sentence how you remove the distortion in the Image

The following sentence has been added on p. 5, l. 8-9:

The distortion of the images is removed with a polynomial function.

(13)     p.6, line 13: could you state a possible explanation for the opposing cloudfree/overcast conditions in winter and summer at Payerne and Davos? Similarly, after line 23, insert a new paragraph and describe the differences in cloud type between the two stations, e.g., fewer Cu and St-As at Payerne with respect to Davos but much more Cb-Ns, most likely due to deficiencies in the cloud type algorithm.

We added a paragraph to further discuss the differences in cloud coverage between the two stations (p. 6, l. 26ff.):

The difference in cloud-free and overcast situations can be explained by the location and the topography of the two stations. In the Midlands, where Payerne is located, in autumn and winter months a common meteorological condition is an inversion, which leads to fog and thus to an overcast sky. Whereas in Davos, located in the Alps, the weather is rather dominated by thermal lift, which occurs more often in summer than in winter.

Another paragraph was added on p. 7, l. 6ff. to further discuss the distribution of cloud types:

In Davos, as determined by our algorithm, from October to May St-As is present in at least 40 % of the cases per month. This fraction of St-As is rather too high and might be due to a limitation of the cloud type algorithm. The limitation is, that the algorithm applied for Davos is trained with images from Payerne. Therefore it might be more difficult to distinguish between low-level cloud classes (e.g. St-As and Sc) in Davos. This limitation might also be responsible for the rather infrequent determination of Cu in Davos.

(14)     **p.7, Fig.2 in the legend: Cb instead of Cn**

Done

(15)     **p.7, line 1. "visual observations": Do you refer to routinely conducted synoptic cloud observations by trained personal, i.e. human observer?**

With visual observations we meant the visual analysis of images and therefore changed the sentence on p. 7, l. 11ff.:

This absence of the cloud class Ci-Cs in the late summer months does not match with the visual analysis of images and might be explained by the fact that the cloud detection algorithm is not sensitive enough for thin high-level clouds.

(16)     **p.7, line 14: to some extent also for St-As.**

Yes the non-linearity is also seen for St-As. Therefore we added St-As as well at p. 8, l. 10-11:

This non-linear increase is clearly observed for the cumulus type clouds Cu, Sc and Cc-Ac, as well as for St-As.

(17)     **p.7, lines 15-20: I would state the statistics for Ci-Cs and 8 oktas coverage for Davos, even if it is too high. Concerning the causes for this particular case, I do not believe that the erroneous values are due to the fact that the camera is not sensitive to high-level clouds. It is not reasonable that the camera detects high-level clouds with lower cloud coverage (these values seem to be reasonable) but does not for overcast conditions. Thus, I would rephrase lines 15-20 which are anyways partly difficult to understand, e.g. something like: "The median for overcast (8 oktas cloud coverage) Ci-Cs conditions in Davos is clearly too high at XX Wm-2. Manually checked images indicate a misclassification of numerous cases as Ci-Cs instead of a cloud type with a lower cloud base and/or optically thicker clouds. Alternatively, the classification as Ci-Cs could be correct, but various cloud types occur at the same time including clouds with a lower cloud base/optically thicker clouds resulting in higher LCE values for Ci-Cs. A possible reason for the misclassification could be that the algorithm is trained with a data set from Payerne." Finally, I would delete lines 20-24.**

Thanks to your comment we changed this paragraph on p. 10, l. 3ff.:

The median LCE value for Ci-Cs in Davos and eight oktas cloud coverage at 53 $Wm^{-2}$ is clearly too high. Manually checked images indicate a misclassification of numerous cases as Ci-Cs instead of a cloud type with a lower cloud base. A possible reason for the misclassification could be that the algorithm is trained with a data set from Payerne. In general, the greater the fractional cloud coverage, the more difficult it becomes to distinguish among cloud types.

(18)     p.7, line 26: It is Table 1 (instead of 2). It would be also helpful to include the absolute or relative numbers of occurrences for the individual cloud classes and cloud coverages (in the same table or in a separate table). Indeed, some results which are not reasonable as discussed before could be also due to a limited number of occurrences for a particular cloud class and cloud cover.

We added the two tables A1 and A2 in the appendix with the absolute numbers of occurrences per cloud class and cloud fraction. Also in the text it has been further discussed, that some differences might be explained by the limited number of cases.

(19)     p. 9, line 10: "at 36 Wm-2" instead of "with 36 Wm-2"

Done

(20)     p. 9, line 13: I would rephrase this sentence, e.g., " The difference of the median LCE values increases with decreasing cloud coverage." or similar.

Done

(21)     p. 9, line 14/15: I would simply write "The difference might be partly due to a higher underestimation of the calculated LW cloud-free irradiances at Payerne." or similar.

The sentence has been changed as suggested.

(22)     p. 9, line 16: "higher" instead of "larger".

Done

(23)     p. 10, line 4: No new paragraph. Continue directly with "Table 2 summarizes..." on line 4 (and it is Table 2 not Table 3).

Done

(24)     p. 10, line 7: Delete "SCErel value".

Done

(25)     p. 11, line 4: "higher" instead of "lower".

Done

(26)     p. 11, line 12: 2x "conditions" instead of "condition".

Done

(27)     p. 11, line 13: delete "part of the shortwave radiation".

Done

(28)     p. 11, line "at" instead of "with" and "range" instead of "region".

Done

(29)     p. 11, lines 4-6: I would rephrase these two sentences, e.g.: "The largest contribution stems from the cloud class Cc-ac at 32 % of the cases, followed by Cu at 27 %, Sc (20 %), St-As (11 %)....."

Done

(30)     p.11, line 7: "negligibly small at 0.2 %".

Done

(31)     p.12, line 14: "... in 8 % of the 126,148 cloud cases, a cloud enhancement of more than 5 % SCErel is observed."

Done

(32)     p.12, line 25: "Schade et al. (2007) showed..."

Done

(33)     p.12, line 10: "...Davos and Payerne are summarized in Table 3 separately." (it is Table 3).

Done

(34)     p.12, line 12: "...the less negative/the more positive the TCE...".

We changed the sentence on p. 15, l. 1-2:

The smaller the cloud coverage is, the less negative the TCE values are.

(35)     p. 12, line 16/p. 13, line 2: "Among other reasons": You may list two or three of them. In addition to the cloud enhancement, the positive values are most likely also due to the relatively large uncertainties in the cloud-free model. In my opinion, this should be stated here.

We added two more sentences on p. 15, l. 2ff.:

Among other reasons, one reason for these positive values with smaller cloud coverages might be the cloud enhancement events as described in section 3.2.2. Another reason might be the uncertainty in the cloud type detection algorithm as well as a larger uncertainty in SCE values the larger the SZA is.

(36)    p. 17, line 11: "increased" instead of "decreased".

Done

(37)    p. 17/18 lines 18-21: I would rephrase this last paragraph (note: the radiation (not energy) budget would be complete if upwelling fluxes were considered) , something like: "The calculations and observations in this study are limited to daylight hours since the hemispherical sky camera operates in the visible wavelength range. However, for climate-monitoring applications cloud observations during day and night are necessary. Therefore, a new observing system (infrared cloud camera) has been developed..."

We shortened and rewrote the whole conclusions. Therefore also this last sentence has been changed.

[revised manuscript text omitted]

35 The current study presents  a study of cloud radiative  effect at the surface depending on cloud fraction and

cloud types at two stations in Switzerland over a time period of 3-5 years. The data and methods (including the description of the algorithms and the models) are described in section 2. The  cloud radiative effect in the longwave and shortwave  ranges at the two stations Davos and Payerne and  sensitivity analyses are presented and discussed in section 3. Conclusions are outlined in section 4.

**2  Data and Methods**

**2.1  Data**

Data are available from two stations in Switzerland. The stations are located at two altitude levels, Payerne, located in the Midlands (46.49°N, 6.56°E, 490 m asl) and Davos, located in the Swiss Alps (46.81°N, 9.84°E, 1,594 m asl). At both
10  of these stations a visible all-sky camera has been installed. The camera type in Payerne is a VIS-J1006, manufactured by Schreder GmbH (www.schreder-cms.com). This camera system consists of a commercial digital camera (Canon Power Shot A60) with a fisheye lens and a glass dome on top to protect the camera from rain and dust. This camera is sensitive in the red-green-blue (RGB) region of the spectrum and takes two images every five minutes with a resolution of $1200 \times 1600$ pixels each. The two images taken, one just after the other one, have different exposure times (1/500 s and 1/1600 s, respectively) but
15  the same fixed aperture of $f$/8.

The camera system in Davos is a Q24M from Mobotix (www.mobotix.com). It is a commercial surveillance camera with a fisheye lens sensitive in the RGB as well. The resolution of the images is the same as that for the camera in Payerne. In Davos, one image is taken every minute with an exposure time of 1/500 s. The Mobotix camera is ventilated and installed on a solar tracker with a shading disk.

20  The radiation data are retrieved from Kipp and Zonen CMP22 pyranometers (shortwave; 0.3 - 3 $\mu$m) and from Kipp and Zonen CG4 pyrgeometers (longwave; 3 - 100 $\mu$m) at both stations. All the instruments are daily cleaned and traceable to the respective standard groups of the World Radiation Center (WRC). The temperature data used in the current study are measured at 2 m height at both stations. The integrated water vapour (IWV) data are based on GPS measurements (*Bevis et al.*, 1992; *Hagemann et al.*, 2003) and retrieved from the STARTWAVE (STudies in Atmospheric Radiative Transfer and Water Vapour
25  Effects) database (*Morland et al.*, 2006). Aerosol optical depth  (AOD) data, used for the shortwave cloud-free model, are retrieved from precision filter radiometers (PFR, manufactured by PMOD/WRC). Ceilometer data for the retrieval of the cloud base height (CBH) are only available in Payerne. At this station a CHM15k ceilometer from Jenoptik (now Lufft Mess- und Regeltechnik GmbH) is installed (*Wiegner and Geiß*, 2012).

For the Davos station  the cloud radiative effect (CRE) has been calculated from August 7, 2013 to April 30,
30  2017 with a time resolution of one minute. Data have only been taken into account for daytime measurements when the sun is located minimum five degrees above the horizon and the mountains. For Payerne, the  study of CRE includes data from January 1, 2013 to April 30, 2017 with a time resolution of five minutes. Data  considered

are during daytime with a solar zenith angle (SZA) of maximum 78°. Cloud camera data availability in these periods is around 98 % and 86 % for Davos and Payerne respectively which mainly results from occasional data gaps of 1 to 3 consecutive days. The lower data availability in Payerne can be explained by two longer time periods of more than 20 consecutive days (one in winter and one in summer) when no camera data are available.

**2.2 Cloud Radiative Effect**

In the current study, the cloud radiative effect (CRE) is defined as a radiation measurement value minus a modelled  cloud-free value. The total cloud radiative effect (TCE) is divided into shortwave cloud radiative effect (SCE) and longwave cloud radiative effect (LCE)

$$TCE = SCE + LCE = DSR_{obs} - DSR_{cfm} + DLR_{obs} - DLR_{cfm} \qquad (1)$$

which are both calculated  by comparing an observed downward radiation measurement (shortwave (SW): $DSR_{obs}$, longwave (LW): $DLR_{obs}$) with a modelled downward radiation value (SW: $DSR_{cfm}$ and LW: $DLR_{cfm}$). For our calculations, only measurements from downward radiation during daytime are taken into account. The atmospheric conditions (namely temperature and IWV) in the models are assumed to be the same under cloudy and cloud-free conditions. In the following, the SCE values are given as relative values ($SCE_{rel}$) and calculated using Eq. 2.

$$SCE_{rel} = SCE / \underline{SCE_{CSM} DSR_{cfm}} * 100\% \qquad (2)$$

where $\underline{SCE_{CSM} DSR_{cfm}}$ is the modelled  cloud-free irradiance value for the corresponding date and time. $SCE_{rel}$ is used due to the fact that different solar zenith angles lead to large differences in the absolute SCE values. Clouds  increase the measured LW radiation at the surface as they emit LW radiation. Shortwave radiation measured at the surface is usually reduced by clouds as they reflect SW radiation back to space.

**2.3  Cloud-free Models**

For the calculation of the cloud radiative effects two  cloud-free models, one for the shortwave and the other one for the longwave  range, are needed. The  cloud-free model for the longwave is an empirical model with input of measured surface temperature and integrated water vapour (IWV) values and a climatology of the atmospheric temperature profile (*Wacker et al.*, 2014). Comparing the LW radiation measurements of the  cloud-free cases, detected in the aforementioned time period, with the LW radiation values of the  cloud-free model gives a mean difference of  $-0.9 \pm 3.9$ Wm$^{-2}$ and  $-0.5 \pm 8.1$ Wm$^{-2}$ for Davos and Payerne respectively. Thus this difference lies within measurement uncertainty as it has also been shown by *Wacker et al.* (2014).

The shortwave  cloud-free model (used in Eq. 2) is a lookup table (LUT) based on radiative transfer model calculations using  LibRadtran (*Mayer and Kylling*, 2005). The input of the model is a standard atmosphere including several measured atmospheric parameters: solar zenith angle (SZA), aerosol conditions ( Angstrom coefficient and aerosol optical depth (AOD), both interpolated over one day) and IWV. The airmass is calculated with the formula presented by *Kasten and Young* (1989). The  LUT is different for the two stations Davos and Payerne, considering a different range of values that might occur. Measured values of IWV, SZA and aerosol content are then interpolated

5 with the  LUT and downward shortwave  cloud-free irradiance values are available for all the single time steps and the corresponding atmospheric conditions. The difference between SW measurement and the  cloud-free model depends on the SZA. The bigger the SZA, the higher the mean difference. In Davos, the mean difference changes from 7.2 ± 20.7 Wm$^{-2}$ ( 0.9 ± 2.6 %)  for data with SZA $< 50°$ to 5.7 ± 14.7 Wm$^{-2}$ ( respectively. 1.1 ± 3.8 %) for data with SZA $> 50°$. In Payerne, the

10 mean difference is 7.3 ± 41.7 Wm$^{-2}$ (1.0 ± 5.2 %) for data with SZA $< 50°$. The mean difference is with 3.3 ± 34.1 Wm$^{-2}$ (0.6 ± 8.9 %) slightly larger for data with SZA from 50 to 78°.

**2.4 Cloud Fraction and Cloud Type Retrievals**

[revised manuscript text omitted]

**3 Results and Discussion**

**3.1  Occurrence of Cloud Fraction and Cloud Types**

20 The data sets for the calculation of the cloud radiative effect (CRE) consist of 595,806 and 117,763 images for Davos and Payerne respectively. In Davos, the cloud coverage is eight oktas for 35 % of the data set. In 17 % of the cases the cloud coverage is zero okta, which means a fractional cloud coverage of maximum 5 %. Seven oktas cloud coverage occurs in 11 % of the cases followed by one okta (10 %). Two to six oktas cloud coverage are all equally distributed in 5 to 6 % of the cases.

25 Also in Payerne, a cloud coverage of eight oktas is determined in most of the cases (41 %), followed by zero okta in 25 % of the cases. In 10 % of the cases a cloud coverage of 1 okta is determined followed by seven oktas (6 % of the cases) and two oktas (5 %). A cloud coverage of three to six oktas is determined in 3 - 4 % of the cases. The distribution of the cloud coverage over the months is shown for Davos and Payerne separately in Figure 1. The colours indicate okta cloud coverages. In the winter half year (with a maximum in March and December) the sky is more often

30 cloud-free than in the summer half year in Davos. In contrast, in May the sky is covered with eight oktas in almost half of the cases. Cloud coverages of 1 to 7 oktas are quite equally distributed over the months. In Payerne the situation is opposite for cloud-free days with more frequent eight oktas cloud coverage in wintertime whereas cloud-free situations are more common during summertime. Also in Payerne, cloud coverages of 1 to 7 oktas are fairly equally distributed.

[Figure]

**Figure 1.** Relative frequencies of cloud coverages in 1 to 8  okta divisions (all cloud types together) for the two stations Davos (left) and Payerne (right).

The difference in cloud-free and overcast situations can be explained by the location and the topography of the two stations. In the Midlands, where Payerne is located, in autumn and winter months a common meteorological condition is an inversion, which leads to fog and thus to an overcast sky. Whereas in Davos, located in the Alps, the weather is rather dominated by thermal lift, which occurs more often in summer than in winter.

5  Regarding the distribution of the cloud coverages in  oktas throughout the day, no real pattern can be observed in Davos. In Payerne there are more cloud-free conditions in the early morning than later in the day. The other  okta cloud coverages are also equally distributed throughout the day.

In Davos, of the  595,806 cases, St-As, with 37 % of the cloud cases, is the cloud type that is most detected in the studied time period. The second and third most detected sky conditions in Davos are Cf and Cc-Ac with 17 % and 14 %

10  respectively, followed by Sc (13 %), Cu (12 %), Ci-Cs (5 %) and Cb-Ns (2 %).

In Payerne, of the  117,763 sky images, in 31 % of the cases the cloud type Sc is detected. This is followed by Cf in around 25 % of cases, Cb-Ns, Cc-Ac  and Ci-Cs (each 11 %), St-As (7 %) and Cu (4 %).

Figure 2 shows the relative frequencies of the cloud classes per month for the two stations Davos and Payerne separately and all cloud coverages together. In Davos, as determined by our algorithm, from October to May St-As is present in at least 40 %

15  of the cases per month.  This fraction of St-As is rather too high and might be due to a limitation of the cloud type

[Figure]

**Figure 2.** Relative frequencies of all cloud classes per month (all cloud coverages together) for the two stations Davos (left) and Payerne (right). Sc: stratocumulus, Cu: cumulus, St-As: stratus-altostratus, Cb-Ns: cumulonimbus-nimbostratus, Cc-Ac: cirrocumulus-altocumulus, Ci-Cs: cirrus-cirrostratus, Cf: cloud-free.

algorithm. The limitation is, that the algorithm applied for Davos is trained with images from Payerne. Therefore it might be more difficult to distinguish between low-level cloud classes (e.g. St-As and Sc) in Davos. This limitation might also be responsible for the rather infrequent determination of Cu in Davos. The cloud class Cc-Ac is more often present in summertime than in wintertime. Ci-Cs is almost absent in the months August to October. This absence of the cloud class Ci-Cs in the late

5 summer months does not match with the visual  analysis of images and might be explained by the fact that the  cloud detection algorithm is not sensitive  enough to detect thin high-level clouds. The largest fraction of cloud type in Payerne is Sc for all months. The cloud classes Cb-Ns and St-As are both more often observed during wintertime than during summertime. The larger frequency of these two cloud types agree with the fact that there is more often fully covered sky in wintertime than summertime.

10 Regarding the distribution of the cloud classes throughout the day, there are no large differences in the occurrence of cloud types per time of day. The distribution is quite flat for both stations.

**3.2 Cloud Radiative Effect**

**3.2.1 Longwave Cloud Effect**

Applying Equation 1, the longwave cloud radiative effect (LCE) is calculated for Davos and Payerne and the six cloud classes separately. The dependence of LCE on fractional cloud cover for the above mentioned time period for all six cloud classes is shown for Davos in Figure 3. The boxplots in the figure show the median (red line), the interquartile range (blue box) and the values that are within 1.5 times the interquartile range of the box edges (black line) per okta cloud coverage.

Figure 3 shows a non-linear increase in the LCE with increasing fractional cloud coverage for some cloud classes. This non-linear increase is clearly observed for the cumulus type clouds Cu, Sc and Cc-Ac, as well as for St-As. Clouds at different zenith angles in the sky have a stronger or weaker impact on the downward longwave radiation measured at the surface. In case the zenith angles of the clouds are not equally distributed in our analysed time period, this might be a reason for this non-linearity in LCE. However, we have not analysed it in more detail yet and is subject of a future study. The cloud classes St-As and Cb-Ns are mainly present with a cloud coverage of 5 oktas and more. The median LCE value for Ci-Cs in Davos and eight oktas cloud coverage at 53 $\mathrm{Wm}^{-2}$ is clearly too high. Manually checked images indicate a misclassification of numerous cases as Ci-Cs instead of a cloud type with a lower cloud base. A possible reason for the misclassification could be that the algorithm is trained with a data set from Payerne. In general, the greater the fractional cloud coverage, the more difficult it becomes to distinguish among cloud types. For the cloud type Cc-Ac there are several LCE values of around 40 $\mathrm{Wm}^{-2}$ and small cloud coverages. These high values are obtained in early mornings when the cloud is located in the vicinity of the horizon.

Table 1 gives an overview of the median values and their interquartile range of the LCE per okta cloud coverage for the six cloud classes for Davos and Payerne separately. The number of cases per cloud class and cloud fraction can be found in the appendix (Table A1 and A2).

In Davos, the highest median LCE for a cloud coverage of 8 oktas is observed for the low-level cloud classes Cb-Ns, St-As, Cu and Sc with a maximum influence on the downward longwave radiation at the surface for Cb-Ns (67 $\mathrm{Wm}^{-2}$). The mid-level and thinner cloud class Cc-Ac has a lower median LCE of 49 $\mathrm{Wm}^{-2}$ for a cloud coverage of 8 oktas. Clearly lower is the median LCE value for the high-level cloud class Ci-Cs and 7 oktas cloud coverage (13 $\mathrm{Wm}^{-2}$). Also for other cloud coverages median LCE values of the three low-level cloud types Sc, Cu and St-As stay in the same range.

Although the numbers differ between the two stations, the same pattern holds also for Payerne, namely that the lower the cloud,

[Figure]

**Figure 3.** Dependence of LCE on cloud coverage for Davos for cloud classes stratocumulus (Sc), cumulus (Cu), stratus-altostratus (St-As), cumulonimbus-nimbostratus (Cb-Ns), cirrocumulus-altocumulus (Cc-Ac) and cirrus-cirrostratus (Ci-Cs). Data points (yellow dots) and box plots per  okta with median (red line), interquartile range (blue box) and spread without outliers.

**Table 1.** Median and interquartile range of longwave cloud radiative effect values [Wm$^{-2}$] per  okta for the two stations Davos (DAV) and Payerne (PAY) and six cloud classes stratocumulus (Sc), cumulus (Cu), stratus-altostratus (St-As), cumulonimbus-nimbostratus (Cb-Ns), cirrocumulus-altocumulus (Cc-Ac) and cirrus-cirrostratus (Ci-Cs).

| cc [okta] | station | Sc [Wm$^{-2}$] | Cu [Wm$^{-2}$] | St-As [Wm$^{-2}$] | Cb-Ns [Wm$^{-2}$] | Cc-Ac [Wm$^{-2}$] | Ci-Cs |
|---|---|---|---|---|---|---|---|
| 1 | DAV |  8 (2,14) | 0 (-2,3) | - (-,-) | - (-,-) | 0 (-3,3) | 1 (- |
| | PAY | 8 (2,13) | 4 (-2,9) | 6 (2,8 | - (-,-) | 4 (-1,9) | 0 (-4 |
| 2 | DAV | 9 (5,15) | 3 (0,6) | 10 (5,14) | - (-,-) | 4 (0,8) | 3 |
| | PAY | 14 (8,22) | 13 (6,21) |  (14,30) |  (-,-) | 13 (5,20) | 4 (0, |
| 3 | DAV | 15 (9,21) | 8 (4,13) | 18 (8,24) | - (-,-) | 5 (1,11) | 4 |
| | PAY | 39 (22,53) | 21 (14,29) | 30 (23,36) |  (-,-) | 18 (10,27) | 7 (3,1 |
| 4 | DAV | 21 (15,29) | 14 (9,20) | 23 (17,28) |  (-,-) | 9 (4,15) | 7 ( |
| | PAY | 36 (25,47) | 26 (19,32) | 38 (31,46) | 66 (51,75) | 23 (15,33) | 10 (5, |
| 5 | DAV | 27 (18,35) | 22 (18,28) | 23 (15,32) | 54 (46,64) | 15 (9,21) | 9 |
| | PAY | 37 (27,47) | 29 (22,34) | 37 (32,49) | 57 (50,68) | 27 (18,37) | 12 (7,1 |
| 6 | DAV | 35 (26,44) | 34 (26,47) | 32 (22,44) | 51 (42,60) | 22 (16,29) | 9 |
| | PAY | 41 (31,52) | 36 (28,44) | 41 (32,64) | 58 (50,66) | 32 (22,42) | 15 (10, |
| 7 | DAV | 48 (39,56) | 57 (50,63) | 47 (33,56) | 56 (48,64) | 32 (24,41) |  |
| | PAY | 47 (36,56) | 57 (354 (33,65) | 65 (50,73) | 57 (49,64) | 36 (28,46) | 17 (12, |
| 8 | DAV | 61 (54,67) | 63 (58,68) | 65 (56,71) | 67 (61,73) | 49 (40,57) | - |
| | PAY | 59 (49,67) | 62 (58,72) | 72 (67,76) | 63 (54,70) | 37 (26,51) | 20 (15, |

the higher the LCE value. Thus for Payerne, the four low-level cloud types (Sc, Cu, St-As and Cb-Ns) and eight  oktas cloud coverages have median LCE values of 59 - 72 Wm$^{-2}$ (with interquartile ranges of maximum ±10 Wm$^{-2}$). The median LCE value for the mid-level cloud class Cc-Ac and eight  oktas cloud coverage is at 37 Wm$^{-2}$ clearly lower than the values for the low-level clouds and also in comparison with the same  value in Davos. The median LCE value for the high-level cloud class Ci-Cs and 8 oktas is around 22 Wm$^{-2}$. This value is only slightly lower for smaller cloud coverages.

The difference of the median LCE values  between the two stations increases with decreasing cloud coverage. Except Sc and Cb-Ns, the LCE values are generally larger for the station Payerne in comparison with Davos.  The difference might be partly due to a higher underestimation of the calculated LW cloud-free irradiances at Payerne. Another explanation for this difference might be that Payerne is located at a lower altitude level and thus the cloud base temperature is higher, which leads to a larger emission of LW radiation. Some of the differences might also occur due to a limited number of cases in the specific groups (see Table A1 and A2). Thus, some of the numbers have to be taken with caution.

**3.2.2 Shortwave Cloud Effect**

**Table 2.** Median and interquartile range of relative shortwave cloud radiative effect values [%] per okta for the two stations Davos (DAV) and Payerne (PAY) and six cloud classes stratocumulus (Sc), cumulus (Cu), stratus-altostratus (St-As), cumulonimbus-nimbostratus (Cb-Ns), cirrocumulus-altocumulus (Cc-Ac) and cirrus-cirrostratus (Ci-Cs).

| cc [okta] | station | Sc [%] | Cu [%] | St-As [%] | Cb-Ns [%] | Cc-A... |
|---|---|---|---|---|---|---|
| 1 | DAV | 4 (1,5) | 1 (1,4) | - (-,-) | - (-,-) | 1 ( |
| 1 | PAY | -6 (-28,5) | 1 (-29,9) | - (-,-) | - (-,-) | 3 (-18 |
| 2 | DAV | 2 (-22,11) | 3 (-5,7) | 10 (6,15) | - (-,-) | 3 (- |
| 2 | PAY | -7 (-37,7) | -13 (-52,12) | -37 (-42,-15) | - (-,-) | -19 |
| 3 | DAV | -4 (-49,13) | 5 (-23,10) | 15 (11,27) | - (-,-) | 3 (-20 |
| 3 | PAY | -55 (-68,-39) | -28 (-56,12) | -32 (-44,-17) | - (-,-) | -29 -3 |
| 4 | DAV | -14 (-51,14) | -5 (-51,12) | 19 (-18,32) | - (-,-) | -3 (-46 |
| 4 | PAY | -60 (-66,-51) | -43 (-59,2) | -42 (-52,-27) | -46 (-72,-37) | -29 ( |
| 5 | DAV | -25 (-53,13) | -44 (-64,4) | -26 (-50,2) | -60 (-72,-43) | -21 (-54,1 |
| 5 | PAY | -54 (-63,-44) | -49 (-61,-23) | -31 (-53,-21) | -54 (-77,-29) | -28 (- |
| 6 | DAV | -38 (-55,-6) | -60 (-70,-48) | -39 (-54,-11) | -63 (-72,-45) | -21 (-51,8 |
| 6 | PAY | -50 (-60,-39) | -42 (-59,-8) | -39 (-62,-20) | -63 (-76,-39) | -24 -2 |
| 7 | DAV | -45 (-58,-26) | -71 (-78,-61) | -45 (-57,-26) | -66 (-78,-52) | -37 (-55, |
| 7 | PAY | -48 (-58,-35) | -59 (-68,-30) | -61 (-71,-46) | -64 (-77,-43) | -25 |
| 8 | DAV | -62 (-72,-49) | -78 (-85,-70) | -62 (-75,-48) | -90 (-95,-82) | -66 (-77,-5 |
| 8 | PAY | -63 (-76,-51) | -66 (-79,-57) | -73 (-79,-65) | -82 (-89,-71) | -48 (-62,-3 |

Table 2 summarizes the median of the SCE$_{rel}$ and the corresponding interquartile range for cloud coverages of one to eight oktas and for the cloud classes for the two stations Davos and Payerne separately. The relative shortwave cloud radiative effect (SCE$_{rel}$) is calculated using Eq. 2. The number of occurrence per cloud class and cloud fraction are shown in Table A1 and A2.

5  In Davos, the cloud type Cb-Ns, with -90 %, is the cloud type with the largest attenuation for eight oktas cloud coverage. The second lowest SCE$_{rel}$ value for eight oktas cloud coverage is observed for the cloud type Cu (-78 %), followed by Cc-Ac (-67 %). The cloud classes St-As and Sc (both -62 %) are almost in the same range. The uncertainty ranges given as interquartile range are for a fully covered sky up to ±14 %. Also here no statistical values have been calculated for the high-level cloud class Ci-Cs and a cloud coverage of 8 oktas due to the

10 same explanation as given in Section 3.2.1. However the median SCE$_{rel}$ for Ci-Cs and 1 to 7 oktas cloud coverage is in comparison to the low-level cloud classes clearly less negative with values between 1 and -9 %. In general, the median SCE$_{rel}$ values become higher the smaller the cloud coverage is. This behaviour is obtained for all cloud classes.

In Payerne, a different order is observed in the lowest to the highest SCE$_{rel}$ values for a cloud coverage of eight oktas. The

15 cloud class with the lowest values, and thus the largest effect on SW radiation, is again Cb-Ns with -82 %, followed by St-As (-73 %), Cu (-66 %) and Sc (-63 %). The interquartile ranges are in a similar range as the ones for Davos. All these four cloud classes are low-level cloud types and also thicker clouds than the ones at a higher level. Therefore it is reasonable to

infer that these are the four cloud classes with the greatest effect on the downward shortwave radiation. For Payerne, a clearly less negative median $SCE_{rel}$ is observed for the mid-level cloud class Cc-Ac and a cloud coverage of eight oktas (-47 %) in comparison to low-level clouds. The highest median $SCE_{rel}$ value for 8 oktas cloud coverage is observed for the high-level cloud class Ci-Cs (-29 %).

5  The differences in $SCE_{rel}$ values between Davos and Payerne are for several cloud types and cloud coverages rather high (e.g. 33 % for Cc-Ac and 3 oktas). An explanation for these larger differences, mainly for smaller cloud coverages, is the so-called cloud enhancement phenomenon, since the positive $SCE_{rel}$ values might increase the median of $SCE_{rel}$. A cloud enhancement phenomenon describes an event where more downward shortwave radiation is measured at the surface under cloudy conditions than expected under cloud-free conditions. Scattering at cloud edges lead to

10  a focusing effect producing a local enhancement of the SW radiation.

For the calculation of the values in Table 2 different numbers of cases have been taken into account (see appendix Table A1 and A2). Analysing e.g. the images that belong to the group St-As and 2 oktas in more detail, leads to the result that at all the 14 images for this specific group in Payerne the sun is covered by a cloud, whereas in Davos, of the 58 images only in around

15  20 % of the cases the sun is occulted and in the remaining 80 % the sun is visible. As further discussed in Section 3.3.2, this fact of visible or occulted sun can lead to a large difference in $SCE_{rel}$ values. These larger differences in $SCE_{rel}$ values between the two stations mainly occur when only a limited number of images is available. Therefore, some of the $SCE_{rel}$ values have to be taken with caution.

[revised manuscript text omitted]

**3.2.3 Total Cloud Effect**

The total cloud radiative effect (TCE) is calculated as the sum of the LCE and SCE (Eq. 1). The calculated median TCE values and the corresponding interquartile range for cloud coverages of one to eight  oktas and the cloud classes for the two stations Davos and Payerne  are summarised in Table 3 separately. For the calculation of TCE, the absolute values of SCE are taken into account and Eq. 2 is not applied. The TCE values are mainly given to get an idea whether the SCE or the LCE is the prevailing contributor to the TCE during daytime.

During daytime, the SCE values are the main contribution to the TCE for all cloud classes and cloud coverages of 6 to 8  oktas and the two stations Davos and Payerne. For the low-level cloud type Cb-Ns, the TCE values are negative for all  oktas cloud coverages. Thus during daytime the SCE is the main contributor to TCE for this cloud class. The smaller the cloud coverage is, the  less negative the TCE values are. This behaviour can be seen for all cloud types and both stations. Among other reasons, one reason for these positive values with smaller cloud coverages might be the cloud enhancement events as described in section 3.2.2. Another reason might be the uncertainty in the cloud type detection algorithm as well as a larger uncertainty in SCE values the larger the SZA is.

[Figure]

**Figure 5.** Dependence of LCE on integrated water vapour (IWV) for Davos and cloud coverage of 8  oktas for low-level clouds (Sc, Cu, St-As, Cb-Ns) shown as a density plot.

**3.3 Sensitivity Analysis**

**3.3.1 Longwave Cloud Effect**

[revised manuscript text omitted]

Our study confirmed, that the cloud base height and the fractional cloud coverage have an influence on the range of the LCE.

10 The higher the cloud coverage, the greater the LCE and the lower the cloud base height, the larger the LCE.

 We also showed that there is a negative dependence of the LCE on integrated water vapour. A similar trend was observed using radiative transfer modelling studies

15 , as well as by *Wacker et al.* (2011).

20  Low-level

clouds have a greater effect on the SCE (up to - 90 % for Cb-Ns) than mid- (up to - 66 %) or high-level clouds (- 28 %). However, not only cloud parameters have an influence, but also whether the sun is  visible or occulted. There are two different distributions depending on whether the measured direct SW radiation exceeds a threshold of 120 $\mathrm{Wm}^{-2}$ or not.

5  : one has its maximum at around - 65 % (occulted sun) and the other one around 0 % ~~SCE$_{\mathrm{rel}}$ for Davos for the cloud type Cc-Ac.This difference is that high because the direct part of the shortwave radiation contributes most to the total shortwave radiation at the surface. Not only the mid-level cloud class shows these two peaks, but also all the low-level cloud classes Cu, Sc, St-As and Cb-Ns individually. For each of these low-level cloud classes the two peaks of the SCE$_{\mathrm{rel}}$ values are also in the same range. The differences in median SCE$_{\mathrm{rel}}$ values between Davos and Payerne are even larger than for the LCE,~~

10 (visible sun).

Our data show that in 14 % and 10 % of the cases in Davos and Payerne respectively a shortwave cloud radiative enhancement of at least 5 % is observed. We  show that Cc-Ac is the cloud type that is responsible for  at least one third of the cloud enhancement cases in Davos and Payerne.

15

In the current analysis, only one cloud type per cloud camera image is defined. A step forward would be to distinguish between different cloud types per image. This detection of different cloud types per image is already an intermediate step in our algorithm. At the current state the cloud type with most of the hits is determined. A further advance would be to not only get the

20 most probable cloud type per image but also to obtain the different cloud types per image as output. Thereafter a more accurate analysis considering the influence of the cloud type on the cloud radiative effect would be possible.

To

25 further minimise the number of misclassifications, for a future study it might be enough to distinguish between low-, mid- and high-level clouds instead of cloud types. This would also increase the number of cases per cloud type and cloud fraction and might decrease the uncertainty of the cloud type detection algorithm. However, it would also decrease the variety in the cloud information.

Another step foreward might be to combine different cloud detection instruments. A new observing system (thermal infrared

30 cloud camera) has been developed in order to collect all-sky cloud information from day- and nighttime measurements. This increase of the data set to nighttime information is necessary for climate-monitoring applications.

[revised manuscript text omitted]